# ROBUSTIFYING AND BOOSTING TRAINING-FREE NEURAL ARCHITECTURE SEARCH

**Zhenfeng He**[1], **Yao Shu**[2,*] **Zhongxiang Dai**[3], **Bryan Kian Hsiang Low**[1]
[1]Department of Computer Science, National University of Singapore
[2]Guangdong Lab of AI and Digital Economy (SZ)
[3]Laboratory for Information and Decision Systems, Massachusetts Institute of Technology
he.zhenfeng@u.nus.edu, shuyao@gml.ac.cn, daizx@mit.edu, lowkh@comp.nus.edu.sg

## ABSTRACT

*Neural architecture search* (NAS) has become a key component of AutoML and a standard tool to automate the design of deep neural networks. Recently, training-free NAS as an emerging paradigm has successfully reduced the search costs of standard training-based NAS by estimating the true architecture performance with only training-free metrics. Nevertheless, the estimation ability of these metrics typically varies across different tasks, making it challenging to achieve robust and consistently good search performance on diverse tasks with only a single training-free metric. Meanwhile, the estimation gap between training-free metrics and the true architecture performances limits training-free NAS to achieve superior performance. To address these challenges, we propose the *robustifying and boosting training-free NAS* (RoBoT) algorithm which *(a)* employs the optimized combination of existing training-free metrics explored from Bayesian optimization to develop a robust and consistently better-performing metric on diverse tasks, and *(b)* applies greedy search, i.e., the exploitation, on the newly developed metric to bridge the aforementioned gap and consequently to boost the search performance of standard training-free NAS further. Remarkably, the expected performance of our RoBoT can be theoretically guaranteed, which improves over the existing training-free NAS under mild conditions with additional interesting insights. Our extensive experiments on various NAS benchmark tasks yield substantial empirical evidence to support our theoretical results. Our code has been made publicly available at `https://github.com/hzf1174/RoBoT`.

## 1 INTRODUCTION

In recent years, deep learning has witnessed tremendous advancements, with applications ranging from computer vision (Caelles et al., 2017) to natural language processing (Vaswani et al., 2017). These advancements have been largely driven by sophisticated deep neural networks (DNNs) like AlexNet (Krizhevsky et al., 2017), VGG (Simonyan & Zisserman, 2014), and ResNet (He et al., 2016), which have been carefully designed and fine-tuned for specific tasks. However, crafting such networks often demands expert knowledge and a significant amount of trial and error. To mitigate this manual labor, the concept of neural architecture search (NAS) was introduced, aiming to automate the process of architecture design. While numerous training-based NAS algorithms have been proposed and have demonstrated impressive performance (Zoph & Le, 2016; Pham et al., 2018), they often require significant computational resources for performance estimation, as it entails training DNNs. More recently, several *training-free* metrics have been proposed to address this issue (Mellor et al., 2021; Abdelfattah et al., 2020; Shu et al., 2021). These metrics are computed with at most a forward and backward pass from a single mini-batch of data, thus the computation cost can be deemed negligible compared with previous NAS methods, hence training-free.

However, there are two questions that still need to be investigated. While training-free metrics have demonstrated promising empirical results in specific, well-studied benchmarks such as NAS-

*Corresponding author.

Bench-201 (Dong & Yang, 2020), recent studies illuminated their inability to maintain consistent performance across diverse tasks (White et al., 2022). Moreover, as there is an estimation gap between the training-free metrics and the true architecture performance, how to quantify and bridge this gap remains an open question. We elaborate on the details of these questions in Section 3.

To address the questions, we propose *robustifying and boosting training-free neural architecture search* (RoBoT). We explain the details of RoBoT in Section 4. First, we use a weighted linear combination of training-free metrics to propose a new metric with better estimation performance, where the weight vectors are explored and optimized using *Bayesian optimization* (BO) (Section 4.1, 4.2). Second, we quantify the estimation gap of the new metric using *Precision value* and then bridge the gap with greedy search. By doing so, we exploit the new metric, hence boosting the expected performance of the proposed neural architecture (Section 4.3).

To understand the expected performance of our proposed algorithm, we make use of the theory from partial monitoring (Bartók et al., 2014; Kirschner et al., 2020) to provide theoretical guarantees for our algorithm (Section 5). To the best of our knowledge, we are the first to provide a bounded expected performance of a NAS algorithm utilizing training-free metrics with mild assumptions. We also discuss the factors that influence the performance. Finally, we conduct extensive experiments to ensure that our proposed algorithm outperforms other NAS algorithms, and achieves competitive or even superior performance compared to any single training-free metric with the same search budget and use ablation studies to verify our claim (Section 6).

## 2 Related Work

**Training-free NAS** Recently, several training-free metrics have emerged, aiming to estimate the generalization performance of neural architectures. This advancement allows NAS to bypass the model training process entirely. For example, Mellor et al. (2021) introduced a heuristic metric that leverages the correlation of activations in an initialized DNN. Similarly, Abdelfattah et al. (2020) established a significant correlation between previously employed training-free metrics in network pruning, such as *snip*, *grasp* and *synflow*. Ning et al. (2021) demonstrated that even simple proxies like *params* and *flops* possess significant predictive capabilities and serve as baselines for training-free metrics. Despite these developments, White et al. (2022) noted that no existing training-free metric consistently outperforms others, emphasizing the necessity for more robust results.

**Hybrid NAS** A few initial works have sought to combine the training-free metrics information and training-based searches. Some methods, such as OMNI (White et al., 2021b) and ProxyBO (Shen et al., 2023), employ a model-based approach, utilizing training-free metrics as auxiliary information to enhance their estimation performance. However, requiring modeling of the search space limits the flexibility when these methods are applied to diverse tasks, and the effectiveness of these methods is questionable when applied to new search space. Moreover, a comprehensive theoretical analysis regarding the influence of training-free metrics on their results is currently absent. Furthermore, Shu et al. (2022b) directly boosts the performance of the gradient-based training-free metric, but their focus is confined to the enhancement of a single training-free metric, and thus is still subject to the instability inherent to individual training-free metrics.

## 3 Observations and Motivations

Although training-free NAS has considerably accelerated the search process of NAS via estimating the architecture performance using only training-free metrics, recent studies have widely uncovered that these metrics fail to perform consistently well on different tasks (White et al., 2022). To illustrate this, we compare the performance of a list of training-free metrics on various tasks in TransNAS-Bench-101-micro (Duan et al., 2021), as shown in Table 1. The results show that no single training-free metric can consistently outperform the others, emphasizing the demand for achieving robust and consistent search results based on training-free metrics. Therefore, it begs the first research question **RQ1**: *Can we robustify existing training-free metrics to make them perform consistently well on diverse tasks?*

The conventional approach to propose a neural architecture using training-free metrics involves selecting the architecture with the highest score. However, this method is potentially limited due

Table 1: Validation ranking of the highest-score architecture for each training-free metric (the value before comma), and the ranking of the best-performing architecture on each training-free metric (the value after comma) on TransNAS-Bench-101-micro (4,096 architectures).

| **Metrics** | Scene | Object | Jigsaw | Segment. | Normal |
|---|---|---|---|---|---|
| grad_norm (Abdelfattah et al., 2020) | 961, 862 | 2470, 1823 | 3915, 1203 | **274**, 738 | 2488, 1447 |
| snip (Lee et al., 2019) | 2285, 714 | 2387, 1472 | 2701, 883 | 951, 682 | 2488, 1581 |
| grasp (Wang et al., 2020) | **372**, 3428 | 3428, 2922 | 2701, 2712 | 2855, 1107 | 665, **680** |
| fisher (Turner et al., 2020) | 2286, 1150 | 2470, 1257 | 2701, 1582 | 3579, 1715 | 4015, 1761 |
| synflow (Tanaka et al., 2020) | 509, 258 | 1661, **50** | 1691, 760 | n/a | n/a |
| jacob_cov (Mellor et al., 2021) | 433, 1207 | 2301, 2418 | 441, 1059 | 592, **323** | **205**, 2435 |
| naswot (Mellor et al., 2021) | 2130, 345 | **1228**, 1466 | 2671, 626 | 709, 1016 | 442, 1375 |
| zen (Lin et al., 2021) | 509,**106** | 1661, 598 | 2552, **430** | 830, 909 | 442, 1002 |
| params | 773, 382 | 2387, 379 | **231**, 494 | 830, 1008 | 310, 1251 |
| flops | 773, 366 | 2387, 427 | **231**, 497 | 830, 997 | 310, 1262 |

to the *estimation gap* in using training-free metrics to estimate the true architecture performance: There could be a superior architecture within the top-ranked segment of the training-free metric that does not necessarily possess the highest score. For instance, the results displayed in Table 1 demonstrate that the highest-score architecture proposed by the *synflow* metric for the *Object* task performs poorly. However, the optimal architecture within the entire search space can be identified by merely conducting further searches among the top 50 architectures. Despite these findings, no existing studies have explored the extent of this potential, or whether allocating a search budget to exploit training-free metrics would yield valuable results. Therefore, our second research question arises: **RQ2**: *After the development of our robust metric, how can we quantify such an estimation gap and propose a method to bridge this gap for further boosted NAS?*

## 4 Our Methodology

To address the aforementioned questions, we will use this section to elaborate on the method RoBoT we propose. Consider the search space as a set of $N$ neural architectures, i.e., $\mathbb{A} = \{\mathcal{A}_i\}_{i=1}^N$, where each architecture $\mathcal{A}$ has a true architecture performance $f(\mathcal{A})$ based on the objective evaluation metric $f$. Without loss of generality, we assume a larger value of $f$ is more desirable. Alternatively, if the entire search space has been assessed by the objective evaluation metric $f$, we can sort their performance and obtain the rankings regarding $f$ as $R_f(\mathcal{A})$, where $R_f(\mathcal{A}) = 1$ indicates $\mathcal{A}$ is the optimal architecture. We also have a set of $M$ training-free metrics $\mathbb{M} = \{\mathcal{M}_i\}_{i=1}^M$, where a larger value of $\mathcal{M}_i(\mathcal{A})$ suggests $\mathcal{A}$ is more favorable by $\mathcal{M}_i$. Notice that we define the search costs as the number of times querying the objective evaluation metric, as the computation costs of training-free metrics are negligible to the objective evaluation metric, given that the latter usually involves the training process of neural architectures. The goal of NAS is to find the optimal architecture with the search costs $T$.

### 4.1 Robustifying Training-free NAS Metric

As discussed in **RQ1**, training-free metrics usually fail to perform consistently well across various tasks. To robustify them, we propose using an ensemble method to combine them, as ensemble methods are widely known as a tool to achieve a consistent performance (Zhou, 2012; Shu et al., 2022a). We choose weighted linear combination as the ensemble method and discuss other choices in Appendix C.3. Formally, we define the weighted linear combination estimation metric with weight vector $\boldsymbol{w}$ as $\mathcal{M}(\mathcal{A}; \boldsymbol{w}) \triangleq \sum_{i=1}^M w_i \mathcal{M}_i(\mathcal{A})$. To find the optimal architecture, we need to identify the most suitable weight vector $\boldsymbol{w}^*$ as

$$\begin{aligned} \boldsymbol{w}^* &= \arg\max\nolimits_{\boldsymbol{w}} f(\mathcal{A}(\boldsymbol{w})), \\ \text{s.t. } \mathcal{A}(\boldsymbol{w}) &\triangleq \arg\max\nolimits_{\mathcal{A} \in \mathbb{A}} \mathcal{M}(\mathcal{A}; \boldsymbol{w}) \ . \end{aligned} \tag{1}$$

Subsequently, we propose $\widetilde{\mathcal{A}}^*_{\mathbb{M}} = \mathcal{A}(\boldsymbol{w}^*)$, which represents the best architecture we can obtain based on the information of training-free metrics $\mathbb{M}$ using the weighted linear combination.

| | Metric 1 | Metric 2 |
|---|---|---|
| Architecture 1 | 0.05 | 0.9 |
| Architecture 2 | **1** | 0 |
| Architecture 3 | 0.04 | 0.95 |
| Architecture 4 | 0 | **1** |

(a)

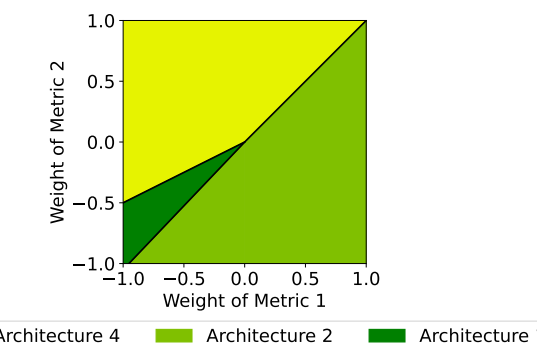

(b)

Figure 1: (a) Training-free Metric 1 and 2's scores for four architectures for a specific task, a higher value indicates more recommended. The true architecture performance is ranked as $1 > 2 > 3 > 4$. (b) The highest-scoring architecture selected based on the weight vector of Metric 1 and Metric 2. Each region of the weight vector selects the same architecture, as represented by its color.

The rationale for opting for a linear combination stems from two key aspects. First, the linear combination has a superior representation ability compared with training-free metrics, where the lower bound can be observed from the one-hot setting of the weight vector (i.e., the weight of the best training-free metric is 1 and the rest are 0). Furthermore, Proposition 1 (prove in Appendix A.1) suggests that a linear combination of two estimation metrics can almost always improve the correlation with the objective evaluation metric, and the exception exists in rare cases such as training-free metrics are co-linear.

**Proposition 1.** *Suppose there are two estimation metrics $\mathcal{M}_1$, $\mathcal{M}_2$, and objective evaluation metric $f$. If $\text{Cov}[\mathcal{M}_1, \mathcal{M}_2] \neq \frac{\text{Cov}[\mathcal{M}_2, f]\text{Var}[\mathcal{M}_1]}{\text{Cov}[\mathcal{M}_1, f]}$ and $\text{Cov}[\mathcal{M}_1, \mathcal{M}_2] \neq \frac{\text{Cov}[\mathcal{M}_1, f]\text{Var}[\mathcal{M}_2]}{\text{Cov}[\mathcal{M}_2, f]}$, $\exists w_1, w_2 \in \mathbb{R}$ such that*

$$\rho_{\textit{Pearson}}(w_1\mathcal{M}_1 + w_2\mathcal{M}_2, f) > \max(\rho_{\textit{Pearson}}(\mathcal{M}_1, f), \rho_{\textit{Pearson}}(\mathcal{M}_2, f))$$

*where $\text{Cov}[X, Y]$, $\rho_{\textit{Pearson}}(X, Y)$ are the covariance and Pearson's correlation between $X$ and $Y$, respectively, and $\text{Var}[X]$ is the variance of $X$.*

This claim can be extended to the scenario of combining multiple metrics, which can be initially treated as a linear combination of two metrics, and then combined with a third. In this context, the strictly increasing properties almost always hold. Hence, Proposition 1 implies that the architecture proposed by a linear combination could outperform any architecture proposed by a single metric.

The second rationale is that linear combinations have sufficient expressive *hypothesis space*, which means improved performance can be obtained from the hypothesis space but it may not be too complex to be optimized. To clarify, the hypothesis space here refers to the possible permutations of architectures given different weight vectors. Given that we may lack a clear understanding of the intricate relationships among training-free metrics, a linear combination serves as a robust and simple choice that contains sufficient good permutation while being easy to optimize.

Although the concept of a linear combination is straightforward, determining the most suitable weight vector can be challenging. These challenges stem from three main factors. First, there might be insufficient prior knowledge regarding the performance of a metric on a specific task. It's common to see inconsistent metric performance across different tasks, as shown in Table 1. Second, even if there is some prior knowledge, the weight vector in the linear combination plays a rather sophisticated role and does more than merely indicate the importance of metrics. For example, Figure 1 shows a scenario where Metric 1 appears to perform better than Metric 2 in a given task (e.g., exhibiting a higher Spearman's rank correlation). However, to choose Architecture 1, which is the optimum for this task, the weight vector must fall within the dark green region, suggesting that Metric 2 generally requires higher weighting than Metric 1, which is the opposite of their performance when used alone. Third, the optimization process is costly, as it involves querying the objective evaluation metric for the true architecture performance.

### 4.2 Exploration to Optimize the Robust Estimation Metric

Considering the aforementioned difficulties, we employ *Bayesian optimization* (BO) (Snoek et al., 2012) to optimize the weight vector. We select BO due to its effectiveness in handling black-box global optimization problems and its efficiency in incurring only small search costs, aligning well with our scenario where the optimization problem is complex and querying is costly. We present our algorithm in Algorithm 1.

**Algorithm 1:** Optimization of Weight Vector through Bayesian Optimization

1: **Input:** Objective evaluation metrics $f$, a set of training-free metrics $\mathbb{M}$, the search space $\mathbb{A}$, and the search budget $T$
2: $\mathcal{Q}_0 = \emptyset$
3: **for** step $t = 1, \ldots, T$ **do**
4:     Update the GP surrogate using $\mathcal{Q}_{t-1}$
5:     Choose $\boldsymbol{w}_t$ using the acquisition function
6:     Obtain the architecture $\mathcal{A}(\boldsymbol{w}_t)$
7:     **if** $\mathcal{A}(\boldsymbol{w}_t)$ is not queried **then**
8:         Query for the objective evaluation metric $f(\mathcal{A}(\boldsymbol{w}_t))$
9:     **else**
10:         Obtain $f(\mathcal{A}(\boldsymbol{w}_t))$ from $\mathcal{Q}_{t-1}$
11:     **end if**
12:     Obtain $\mathcal{Q}_t = \mathcal{Q}_{t-1} \cup \{(\boldsymbol{w}_t, f(\mathcal{A}(\boldsymbol{w}_t)))\}$
13: **end for**
14: Select the best-queried weight vector $\widetilde{\boldsymbol{w}}^* = \arg\max_{\boldsymbol{w}} \{f(\mathcal{A}(\boldsymbol{w})), (\boldsymbol{w}, f(\mathcal{A}(\boldsymbol{w}))) \in \mathcal{Q}_T\}$.

Specifically, in every iteration, we first use a set of observations to update the Gaussian Process (GP) surrogate, which is used to model the prior and posterior distribution based on the observations (line 4 of Algorithm 1). An observation is defined as a pair of the weight vector and the true architecture performance of the corresponding architecture selected based on the weight vector, as defined in Equation 1. Then we choose the next weight vector based on the acquisition function and obtain the corresponding architecture (lines 5-6 of Algorithm 1). If the true architecture performance of this architecture has not been queried yet, we will query it. Otherwise, we obtain it from the previous observations (lines 7-11 of Algorithm 1). We update the observation set and when iterations are complete, we select the best-queried weight vector based on the performance of the corresponding architecture (see Appendix B.2 for more implementation details such as the utilized training-free metrics). With this weight vector, we answer **RQ1** by proposing the robust estimation metric $\mathcal{M}(\cdot; \widetilde{\boldsymbol{w}}^*)$.

### 4.3 Exploitation to Bridge the Estimation Gap

As discussed in **RQ2**, the estimation gap limits the metrics to propose a superior architecture. This estimation gap also exists in our proposed robust estimation metric.

To bridge this estimation gap, we first need to quantify it. We propose to use Precision @ $T$ value, which measures the ratio of *relevant* architectures within the top $T$ architectures based on the estimation metric. Here, an architecture is deemed relevant if it ranks among the top $T$ true-architecture-performance architectures. Formally, with an estimation metric $\mathcal{M}$, Precision @$T$ is defined as

$$\rho_{\mathrm{T}}(\mathcal{M}, f) \triangleq \frac{|\{\mathcal{A}, R_{\mathcal{M}}(\mathcal{A}) \leq T \wedge R_f(\mathcal{A}) \leq T\}|}{T}. \tag{2}$$

The reason for adopting Precision @ $T$ value rather than the more conventionally used measurement such as Spearman's rank correlation stems from two factors. First, it focuses on the estimation of top-performing architectures rather than the entire search space, which aligns with the goal of finding optimal architecture in NAS. Second, if we employ *greedy search* on the top $T$ architectures based on the estimation metric $\mathcal{M}$, and propose the architecture

$$\mathcal{A}^*_{\mathcal{M},T} = \arg\max_{\mathcal{A} \in \mathbb{A}} \{f(\mathcal{A}), R_{\mathcal{M}}(\mathcal{A}) \leq T\}, \tag{3}$$

then we can directly obtain the expected ranking performance of this architecture, as demonstrated in Theorem 1 (the proof is given in Appendix A.2 with the discussion about the uniform distribution assumption of $\mathbb{P}[R_f(A) = t] = 1/T$). According to this theorem, a higher Precision @ $T$ value leads to a better ranking expectation. Hence, using Precision @ $T$ can provide a direct analysis of the performance of the proposed architecture.

**Theorem 1.** *Given the estimation metric $\mathcal{M}$ and the objective evaluation metric $f$, define $\mathbb{A}_{T,\mathcal{M},f} = \{\mathcal{A}, R_{\mathcal{M}}(\mathcal{A}) \leq T \wedge R_f(\mathcal{A}) \leq T\}$. Suppose that $\forall \mathcal{A} \in \mathbb{A}_{T,\mathcal{M},f}, \forall t \in \{1, 2, \cdots, T\}, \mathbb{P}[R_f(\mathcal{A}) =$*

**Algorithm 2:** Robustifying and Boosting Training-Free Neural Architecture Search (RoBoT)

1: **Input:** Objective evaluation metrics $f$, a set of training-free metrics $\mathbb{M}$, the search space $\mathbb{A}$, and the search budget $T$
2: Execute Algorithm 1, and obtain $\widetilde{\boldsymbol{w}}^*$, $T_0 = |\{\mathcal{A}(\boldsymbol{w}), (\boldsymbol{w}, f(\mathcal{A}(\boldsymbol{w}))) \in \mathcal{Q}_T\}|$
3: Obtain robust estimation metric $\mathcal{M}(\cdot; \widetilde{\boldsymbol{w}}^*)$ and corresponding ranking $R_{\mathcal{M}(\cdot;\widetilde{\boldsymbol{w}}^*)}$
4: Propose architecture $\widetilde{\mathcal{A}}^*_{\mathbb{M},T} = \arg\max_{\mathcal{A}\in\mathbb{A}} \{f(\mathcal{A}), \mathcal{A}^*_{\mathcal{M}(\cdot;\widetilde{\boldsymbol{w}}^*),T-T_0} \vee \mathcal{A} \in \mathcal{Q}_T\}$

$t] = 1/T$. *If* $\rho_T(\mathcal{M}, f) \neq 0$, *then*

$$\mathbb{E}[R_f(\mathcal{A}^*_{\mathcal{M},T})] = \frac{T+1}{\rho_T(\mathcal{M}, f)T + 1} \ .$$

Therefore, we employ the greedy search (i.e., exploitation) on the robust estimation metric to bridge the estimation gap. Notably, within Algorithm 1, we may not utilize the entire $T$ search budget (as indicated in lines 9-10). Suppose that there are $T_0$ distinct architectures in $\mathcal{Q}_T$ (corresponding to the number of executions of line 8), this leaves a remaining search budget of $T - T_0$ for exploitation. In practice, the magnitude of $T - T_0$ is usually significant compared with $T$, as shown by our experiments in Section 6. Thus, we apply the greedy search among the top $T - T_0$ architectures of the robust estimation metric to propose the architecture $\mathcal{A}^*_{\mathcal{M}(\cdot;\widetilde{\boldsymbol{w}}^*),T-T_0}$.

By integrating this strategy with Algorithm 1, we formulate our proposed method, referred to as *robustifying and boosting training-free neural architecture search* (RoBoT), as detailed in Algorithm 2. This innovative method explores the weight vector hypothesis space to create a robust estimation metric and exploits the robust estimation metric to further boost the performance of the proposed architecture, thereby proposing a robust and competitive architecture that potentially surpasses those proposed by any single training-free metric, a claim we will substantiate later.

## 5 Discussion and Theoretical Analyses

Our proposed algorithm RoBoT successfully tackles the research questions **RQ1** and **RQ2**. Yet, the assessment of our proposed architecture $\widetilde{\mathcal{A}}^*_{\mathbb{M},T}$ is still open-ended as the value of $\rho_T(\mathcal{M}(\cdot, \widetilde{\boldsymbol{w}}^*), f)$ remains uncertain. To address this, we incorporate our algorithm into the *partial monitoring* framework and provide an analytical evaluation. Based on these findings, we draw intriguing insights into the factors influencing the performance of RoBoT.

### 5.1 Expected Performance of RoBoT

To evaluate Precision @ $T$ value, we note that our Algorithm 1 fits within the *partial monitoring* framework (Bartók et al., 2014). Partial monitoring is a sequential decision-making game where the learner selects an action, incurs a *reward*, and receives an *observation*. The observation is related to the reward, but there is a gap between them. In our context, the action is the selected weight vector, the observation is the true architecture performance of selected architecture and the reward is the Precision @ $T$ between the objective evaluation metric and the robust estimation metric. The goal of a partial monitoring game is to minimize the *cumulative regret*, where in our case it is defined as

$$Reg_t \triangleq \textstyle\sum_{\tau=1}^{t} \max_{\boldsymbol{w}} \rho_{\mathrm{T}}(\mathcal{M}(\cdot; \boldsymbol{w}), f) - \rho_{\mathrm{T}}(\mathcal{M}(\cdot; \boldsymbol{w}_\tau), f). \tag{4}$$

If the cumulative regret is bounded, the expectation of $\rho_{\mathrm{T}}(\mathcal{M}(\cdot, \widetilde{\boldsymbol{w}}^*), f)$ will also be bounded. To quantify the usefulness of the observation, a condition known as *global observability* is used. A global observable game is one in which the learner can access actions that allow estimation of reward differences between different actions based on the observation. If this condition is met, a sublinear cumulative regret $Reg_t \leq \widetilde{\mathcal{O}}(t^{2/3})$ can be achieved (Bartók et al., 2014). In our scenario, we determine that if the conditions of **Expressiveness of the Hypothesis Space** and **Predictable Ranking through Performance** are satisfied, our setup can be considered globally observable (see Appendix A.3 for elaboration). As stated in Kirschner et al. (2020), the partial monitoring can be

extended to reproducing kernel Hilbert Spaces (RKHS) which contains kernelized bandits (i.e., BO). If the above-mentioned conditions can be satisfied, and we use an *information directed sampling* (IDS) strategy as our acquisition function in BO, our Algorithm 1 can achieve a bounded regret $Reg_t \leq \widetilde{\mathcal{O}}(t^{2/3})$. Combine the above brings us to our analysis of $\mathbb{E}[R_f(\widetilde{\mathcal{A}}^*_{\mathbb{M},T})]$.

**Theorem 2.** *Suppose the maximum Precision@$T$ can be achieved by weighted linear combination is $\rho^*_T(\mathcal{M}_{\mathbb{M}}, f) \triangleq \max_{\boldsymbol{w}} \rho_T(\mathcal{M}(\cdot; \boldsymbol{w}), f)$, then for $\widetilde{\mathcal{A}}^*_{\mathbb{M},T}$ obtained from Algorithm 2,*

$$\mathbb{E}[R_f(\widetilde{\mathcal{A}}^*_{\mathbb{M},T})] \leq \frac{T+1}{(\rho^*_T(\mathcal{M}_{\mathbb{M}}, f) - q_T T^{-1/3})(T - T_0) + 1}, \tag{5}$$

*with probability arbitrarily close to 1 when $\rho^*_T(\mathcal{M}_{\mathbb{M}}, f) - q_T T^{-1/3} > 0$. The value of $q_T$ is fixed when $T$ is fixed and $q_T$ depends only logarithmically on $T$.*

Its proof is given in Appendix A.3. With this, we have presented a bounded expectation estimation of the ranking performance of our proposed architecture. In the following section, we will delve into the insights and practical implications derived from Theorem 2.

### 5.2 Discussion and Analysis on RoBoT

With the information provided in Theorem 2, we would like to discuss some factors that influence the performance of the proposed architecture $\widetilde{\mathcal{A}}^*_{\mathbb{M},T}$.

**Influence of $T_0$** As described in Section 4.3, the value of $T_0$ is determined by the number of distinct architectures queried during Algorithm 1. However, under this setting, it implies the value of $T_0$ is not predictable before we perform BO, and is fixed after we perform BO. To analyze the influence of $T_0$, we consider an alternative setting in which we strictly require the BO to run for $T_0$ rounds (and waste the search budget if there are duplicate queries for the same architecture), then conduct exploitation for $T - T_0$ rounds. Under this setting, the denominator in Equation 5 will be updated as $(\rho^*_{\mathrm{T}}(\mathcal{M}_{\mathbb{M}}, f) - q_{T_0} {T_0}^{-1/3})(T - T_0) + 1$.

There are two interesting insights we would like to bring attention. First, as $T_0 \leq T$, for a fixed value of $T_0$, the updated ranking expectation is always worse than the original. Second, as the left and the right terms of the denominator are both influenced by $T_0$, $T_0$ serves as an **explore-exploitation trade-off**. While carefully choosing a $T_0$ might bring a better ranking expectation, our analysis and later ablation study (see Appendix C.3) suggests that keeping $T_0$ to be **automatically** decided as defined in Algorithm 2 would yield a better and more robust performance.

**Influence of $T$** Interestingly, we find that if there is more search budget provided, our proposed architecture is more likely to outperform the architecture proposed by any single training-free metric even if they are given the same time budget, i.e., it is more likely $\mathbb{E}[R_f(\widetilde{\mathcal{A}}^*_{\mathbb{M},T})] \leq \min_{\mathcal{M} \in \mathbb{M}}(\mathbb{E}[R_f(\mathcal{A}^*_{\mathcal{M},T})])$ when $T$ increases (see Appendix A.4 for derivation). This suggests that there will be more advantage to use RoBoT compared with original training-free metrics when more search budget is provided. This claim can be verified in our experiments in Section 6.

## 6 Experiments

### 6.1 RoBoT on NAS benchmark

To empirically verify the robust predictive performance of our proposed RoBoT algorithm across various tasks, we conduct comprehensive experiments on three NAS benchmarks comprising 20 tasks: NAS-Bench-201 (Dong & Yang, 2020), TransNAS-Bench-101 (Duan et al., 2021) and DARTS search space (Liu et al., 2019). Detailed experimental information can be found in Appendix B. Partial results are summarized in Table 2, Table 3, and Table 4, with additional illustrations provided in Figure 2 to depict RoBoT's performance for different numbers of searched architectures. Full results can be found in Appendix C. We clarify how we report the search costs in Appendix B.2.

The results suggest that our proposed architecture consistently demonstrates strong performance across various tasks, achieving at least a similar performance level as the best training-free metric

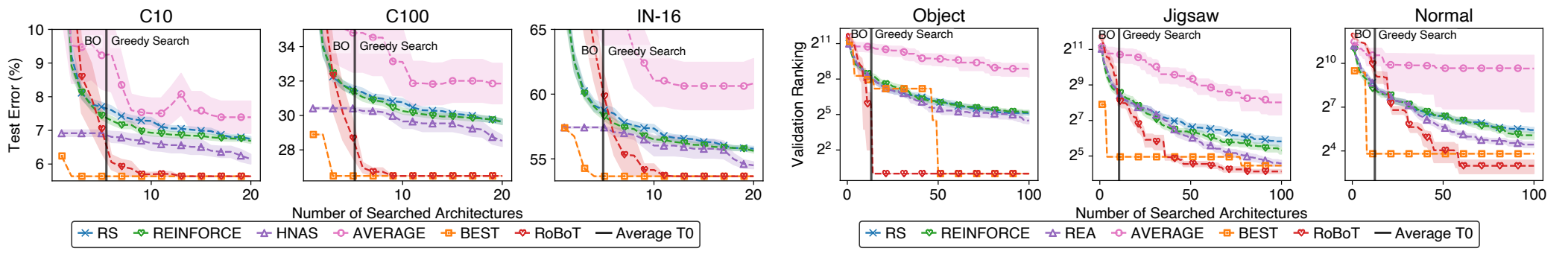

(a) NAS-Bench-201 (b) TransNAS-Bench-101-micro

Figure 2: Comparison of NAS algorithms in NAS-Bench-201 and TransNAS-Bench-101-micro regarding the number of searched architectures. RoBoT and HNAS are reported with the mean and standard error of 10 runs, and 50 runs for RS, REA and REINFORCE.

Table 2: Comparison of NAS algorithms in NAS-Bench-201. The result of RoBoT is reported with the mean ± standard deviation of 10 runs and search costs are evaluated on an Nvidia 1080Ti.

| **Algorithm** | **Test Accuracy (%)** | | | **Cost** (GPU Sec.) | **Method** |
|---|---|---|---|---|---|
| | C10 | C100 | IN-16 | | |
| ResNet (He et al., 2016) | 93.97 | 70.86 | 43.63 | - | manual |
| REA† | 93.92±0.30 | 71.84±0.99 | 45.15±0.89 | 12000 | evolution |
| RS (w/o sharing)† | 93.70±0.36 | 71.04±1.07 | 44.57±1.25 | 12000 | random |
| REINFORCE† | 93.85±0.37 | 71.71±1.09 | 45.24±1.18 | 12000 | RL |
| BOHB† | 93.61±0.52 | 70.85±1.28 | 44.42±1.49 | 12000 | BO+bandit |
| DARTS (2nd) (Liu et al., 2019) | 54.30±0.00 | 15.61±0.00 | 16.32±0.00 | 43277 | gradient |
| GDAS (Dong & Yang, 2019) | 93.44±0.06 | 70.61±0.21 | 42.23±0.25 | 8640 | gradient |
| DrNAS (Chen et al., 2021b) | 93.98±0.58 | 72.31±1.70 | 44.02±3.24 | 14887 | gradient |
| Shaply-NAS (Xiao et al., 2022) | 94.05±0.19 | 73.15±0.26 | 46.25±0.25 | 14762 | gradient |
| $\beta$-DARTS (Ye et al., 2022) | 94.00±0.22 | 72.91±0.43 | 46.20±0.38 | 3280 | gradient |
| NASWOT (Mellor et al., 2021) | 92.96±0.81 | 69.98±1.22 | 44.44±2.10 | 306 | training-free |
| TE-NAS (Chen et al., 2021a) | 93.90±0.47 | 71.24±0.56 | 42.38±0.46 | 1558 | training-free |
| NASI (Shu et al., 2021) | 93.55±0.10 | 71.20±0.14 | 44.84±1.41 | 120 | training-free |
| GradSign (Zhang & Jia, 2022) | 93.31±0.47 | 70.33±1.28 | 42.42±2.81 | - | training-free |
| HNAS (Shu et al., 2022b) | 94.04±0.21 | 71.75±1.04 | 45.91±0.88 | 3010 | hybrid |
| Training-Free | | | | | |
| ↪ Avg. | 91.71±2.37 | 66.48±4.94 | 35.46±9.90 | 2648 | enumeration |
| ↪ Best | **94.36** | **73.51** | **46.34** | 3702 | enumeration |
| RoBoT | **94.36**±0.00 | **73.51**±0.00 | **46.34**±0.00 | 3051 | hybrid |
| **Optimal** | 94.37 | 73.51 | 47.31 | - | - |

† Reported by Dong & Yang (2020).

for nearly all tasks, while the latter is an *oracle* algorithm in practice as it requires to enumerate every training-free metric. Furthermore, RoBoT even outperforms this oracle in several tasks (noting that the architecture proposed by the best training-free metric can already attain near-optimal performance), which exemplifies RoBoT's potential to boost the training-free metrics rather than just ensuring robustness. As illustrated in Figure 2, comparing RoBoT's performance with that of the best training-free metrics reveals that although RoBoT may initially perform worse, it surpasses the best after querying more architectures. This observation aligns with our earlier claim in Section 5.2 regarding the influence of $T$. Moreover, RoBoT generally uses fewer search costs to achieve a similar performance level, which demonstrates its efficiency. Besides, results on DARTS search space presented in Table 4 verifies that RoBoT works well on larger scale search space and practical application as it searches in a pool of 60,000 architectures (please refer to Appendix C.2 for more details). Overall, our findings suggest that RoBoT is an appropriate choice for ensuring the robustness of training-free metrics while having the potential to boost performance.

## 6.2 Ablation Studies

To substantiate our theoretical discussion and bring additional insights regarding RoBoT, we conduct a series of ablation studies to examine factors influencing performance. These factors include optimized linear combination weights, Precision @ $T$, $T_0$, ensemble methods, choice of training-free metrics, and choice of observation. The results and discussion are in Appendix C.3. Overall,

Table 3: Comparison of NAS algorithms in TransNAS-Bench-101-micro (4,096 architectures) and macro (3,256 architectures) regarding validation ranking. All methods search for 100 architectures. The results of RoBoT are reported with the mean ± standard deviation of 10 runs.

| Space | Algorithm | Scene | Object | Jigsaw | Layout | Segment. | Normal | Autoenco. |
|---|---|---|---|---|---|---|---|---|
| Micro | REA | 26±22 | 23±28 | 24±22 | 28±23 | 22±28 | 22±20 | **18**±16 |
| | RS (w/o sharing) | 40±35 | 34±32 | 57±67 | 42±43 | 41±40 | 43±41 | 49±51 |
| | REINFORCE | 35±28 | 37±31 | 40±37 | 38±32 | 34±35 | 33±40 | 34±32 |
| | HNAS | 96±35 | 147±0 | 67±49 | 480±520 | 4±0 | 22±0 | 1391±0 |
| | Training-Free | | | | | | | |
| | ↪ Avg. | 75±145 | 441±382 | 260±244 | 325±318 | 718±1417 | 802±1563 | 1698±125 |
| | ↪ Best | **2** | **1** | 22 | 18 | **3** | 14 | 36 |
| | RoBoT | **2**±0 | **1**±0 | **17**±5 | **17**±34 | 4±1 | **8**±8 | 66±70 |
| Macro | REA | 23±24 | 22±21 | 21±16 | 21 ±20 | 17±20 | 23±22 | 19±17 |
| | RS (w/o sharing) | 35±32 | 26±24 | 33±39 | 29±29 | 33±38 | 26±23 | 31±26 |
| | REINFORCE | 49±45 | 37±31 | 20±22 | 27±25 | 51±43 | 48±39 | 32±17 |
| | HNAS | 552±0 | 566±0 | 150±52 | 49±6 | 16±0 | 204±0 | 674±0 |
| | Training-Free | | | | | | | |
| | ↪ Avg. | 346±248 | 291±275 | 110±77 | 53±25 | 32±23 | 7±4 | 24±22 |
| | ↪ Best | **1** | 8 | 5 | **2** | **2** | **2** | **5** |
| | RoBoT | **1**±0 | **6**±3 | **3**±1 | 6±6 | **2**±2 | **2**±0 | **5**±1 |

Table 4: Performance comparison among SOTA image classifiers on ImageNet on DARTS search space. The search costs are evaluated on an Nvidia 1080Ti.

| **Algorithm** | **Test Error (%)** | | **Params** (M) | $+\times$ (M) | **Search Cost** (GPU Days) |
|---|---|---|---|---|---|
| | Top-1 | Top-5 | | | |
| Inception-v1 (Szegedy et al., 2015) | 30.1 | 10.1 | 6.6 | 1448 | - |
| MobileNet (Howard et al., 2017) | 29.4 | 10.5 | 4.2 | 569 | - |
| NASNet-A (Zoph et al., 2018) | 26.0 | 8.4 | 5.3 | 564 | 2000 |
| AmoebaNet-A (Real et al., 2019) | 25.5 | 8.0 | 5.1 | 555 | 3150 |
| PNAS (Liu et al., 2018) | 25.8 | 8.1 | 5.1 | 588 | 225 |
| MnasNet-92 (Tan et al., 2019) | 25.2 | 8.0 | 4.4 | 388 | - |
| DARTS (Liu et al., 2019) | 26.7 | 8.7 | 4.7 | 574 | 4.0 |
| GDAS (Dong & Yang, 2019) | 26.0 | 8.5 | 5.3 | 581 | 0.21 |
| ProxylessNAS (Cai et al., 2019) | 24.9 | 7.5 | 7.1 | 465 | 8.3 |
| SDARTS-ADV (Chen & Hsieh, 2020) | 25.2 | 7.8 | 5.4 | 594 | 1.3 |
| TE-NAS (Chen et al., 2021a) | 24.5 | 7.5 | 5.4 | - | 0.17 |
| NASI-ADA (Shu et al., 2021) | 25.0 | 7.8 | 4.9 | 559 | 0.01 |
| HNAS Shu et al. (2022b) | 24.3 | 7.4 | 5.1 | 575 | 0.5 |
| RoBoT | **24.1** | **7.3** | 5.0 | 556 | 0.6 |

the ablation studies validate our prior discussions and theorem, providing a more profound comprehension of how these factors affect the performance of RoBoT.

## 7 Conclusion

This paper introduces a novel NAS algorithm, RoBoT, which *(a)* ensures the robustness of existing training-free metrics via a weighted linear combination, where the weight vector is explored and optimized using BO, and *(b)* harnesses the potential of the robust estimation metric to bridge the estimation gap, thus further boosting the expected performance of the proposed neural architecture. Notably, our analyses can be extended to broader applications. For instance, estimation metrics can be expanded to early stopping performance (Falkner et al., 2018; Zhou et al., 2020), and neural architectures can be generalized to a broader set of ranking candidates. We anticipate that our analysis will shed light on the inherent ranking performance of NAS algorithms and inspire the community to further explore the ensemble of training-free metrics, unlocking their untapped potential.

## REPRODUCIBILITY STATEMENT

We have included the necessary details to ensure the reproducibility of our theoretical and empirical results. For our theoretical results, we state all our assumptions in Section 5.1, and provide detailed proof in Appendix A. For our empirical results, the detailed experimental settings have been described in Appendix B and Appendix C.2. Our code has been submitted as supplementary material.

## ACKNOWLEDGEMENT

This research is supported by the National Research Foundation (NRF), Prime Minister's Office, Singapore under its Campus for Research Excellence and Technological Enterprise (CREATE) programme. The Mens, Manus, and Machina (M3S) is an interdisciplinary research group (IRG) of the Singapore MIT Alliance for Research and Technology (SMART) centre.

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

# APPENDIX A PROOFS

## A.1 PROOF OF PROPOSITION 1

Given the definition of Pearson correlation $\rho_{\text{Pearson}}(X,Y) = \frac{\text{Cov}[X,Y]}{\sqrt{\text{Var}[X]\text{Var}[Y]}}$, we aim to maximize the Pearson correlation between the linear combination of two estimation metrics $\mathcal{M}_1, \mathcal{M}_2$ and the objective evaluation metric $f$ over $w_1$ and $w_2$. For the weights $w_1, w_2 \in \mathbb{R}$, the Pearson correlation is:

$$\begin{aligned}
\rho_{\text{Pearson}}(w_1\mathcal{M}_1 + w_2\mathcal{M}_2, f) &= \frac{\text{Cov}[w_1\mathcal{M}_1 + w_2\mathcal{M}_2, f]}{\sqrt{\text{Var}[w_1\mathcal{M}_1 + w_2\mathcal{M}_2]\text{Var}[f]}} \\
&= \frac{\text{Cov}[w_1\mathcal{M}_1, f] + \text{Cov}[w_2\mathcal{M}_2, f]}{\sqrt{(\text{Var}[w_1\mathcal{M}_1] + \text{Var}[w_2\mathcal{M}_2] + 2\text{Cov}[w_1\mathcal{M}_1, w_2\mathcal{M}_2])\text{Var}[f]}} \\
&= \frac{w_1\text{Cov}[\mathcal{M}_1, f] + w_2\text{Cov}[\mathcal{M}_2, f]}{\sqrt{(w_1^2\text{Var}[\mathcal{M}_1] + w_2^2\text{Var}[\mathcal{M}_2] + 2w_1w_2\text{Cov}[\mathcal{M}_1, \mathcal{M}_2])\text{Var}[f]}}.
\end{aligned}$$

Notice that for Pearson correlation, $\forall a > 0, \rho_{\text{Pearson}}(aX, Y) = \rho_{\text{Pearson}}(X, Y)$, and $\forall a < 0, \rho_{\text{Pearson}}(aX, Y) = -\rho_{\text{Pearson}}(X, Y)$. So if $w_2 \neq 0$, $\exists a \in \mathbb{R}, w1 = aw2$, hence $\max(\rho_{\text{Pearson}}(w_1\mathcal{M}_1 + w_2\mathcal{M}_2, f)) = \max(\rho_{\text{Pearson}}(a\mathcal{M}_1 + \mathcal{M}_2, f), -\rho_{\text{Pearson}}(a\mathcal{M}_1 + \mathcal{M}_2, f), \rho_{\text{Pearson}}(\mathcal{M}_1, f))$.

To find out the maximum value of $\pm\rho_{\text{Pearson}}(a\mathcal{M}_1 + \mathcal{M}_2, f)$, we take the derivative of $\pm\rho_{\text{Pearson}}(a\mathcal{M}_1 + \mathcal{M}_2, f)$ regarding $a$ and set it to 0, both yield

$$a = \frac{\text{Cov}[\mathcal{M}_2, f]\text{Cov}[\mathcal{M}_1, \mathcal{M}_2] - \text{Cov}[\mathcal{M}_1, f]\text{Var}[\mathcal{M}_2]}{\text{Cov}[\mathcal{M}_1, f]\text{Cov}[\mathcal{M}_1, \mathcal{M}_2] - \text{Cov}[\mathcal{M}_2, f]\text{Var}[\mathcal{M}_1]}. \tag{6}$$

If $\text{Cov}[\mathcal{M}_2, f]\text{Cov}[\mathcal{M}_1, \mathcal{M}_2] \neq \text{Cov}[\mathcal{M}_1, f]\text{Var}[\mathcal{M}_2]$ and $\text{Cov}[\mathcal{M}_1, f]\text{Cov}[\mathcal{M}_1, \mathcal{M}_2] \neq \text{Cov}[\mathcal{M}_2, f]\text{Var}[\mathcal{M}_1]$, then the solution given in equation 6 exists and it is non-zero. Since $\pm\rho_{\text{Pearson}}(a\mathcal{M}_1 + \mathcal{M}_2, f)$ only has one critical point and the value of $\pm\rho_{\text{Pearson}}(a\mathcal{M}_1 + \mathcal{M}_2, f)$ is bounded within $[-1, 1]$ (as value of Pearson correlation is bounded within $[-1, 1]$), equation 6 must correspond to a global optimum for both $\pm\rho_{\text{Pearson}}(a\mathcal{M}_1 + \mathcal{M}_2, f)$. As $\rho_{\text{Pearson}}(a\mathcal{M}_1 + \mathcal{M}_2, f) = -\rho_{\text{Pearson}}(-(a\mathcal{M}_1 + \mathcal{M}_2), f)$, equation 6 must be a global maximum point for one of $\pm\rho_{\text{Pearson}}(a\mathcal{M}_1 + \mathcal{M}_2, f)$, and the global minimum point for the other.

When the global maximum point exists and its value does not equal either $\rho_{\text{Pearson}}(\mathcal{M}_1, f)$ or $\rho_{\text{Pearson}}(\mathcal{M}_2, f)$, it implies that the value will be strictly larger than both $\rho_{\text{Pearson}}(\mathcal{M}_1, f)$ and $\rho_{\text{Pearson}}(\mathcal{M}_2, f)$, given the definition of global maximum. This concludes the proof.

## A.2 PROOF OF THEOREM 1

With the uniform distribution assumption, our Theorem 1 can be derived directly from the following classical lemma.

**Lemma 1.** *Suppose there are $n$ alternatives with distinct ranking $1, 2, \cdots, n$, and $m \geq 1$ alternatives $\{x_i\}_{i=1}^m$ are uniformly randomly selected from the $n$ alternatives. Suppose $R(x_i)$ is the ranking value of $x_i$, then*

$$\mathbb{E}[\min(\{R(x_i)\}_{i=1}^m)] = \frac{n+1}{m+1}.$$

*Proof.* Let's first consider the probability that $\mathbb{P}[\min(\{R(x_i)\}_{i=1}^m) = k]$ where $k \in \mathbb{Z} \wedge 1 \leq k \leq n - m + 1$. It should be noted that the lowest ranking value should be at most $n - m + 1$ as there are $m$ alternatives with distinct rankings. Given that the lowest value of ranking is $k$, the remaining $m - 1$ alternatives can be selected from those alternatives with ranking values higher than $k$. Thus we have:

$$\mathbb{P}[\min(\{R(x_i)\}_{i=1}^m) = k] = \frac{\binom{n-k}{m-1}}{\binom{n}{m}} \tag{7}$$

The expectation of the lowest ranking value can be derived as

$$
\begin{aligned}
\mathbb{E}[\min(\{R(x_i)\}_{i=1}^m)] &= \sum_{k=1}^{n-m+1} k\mathbb{P}[\min(\{R(x_i)\}_{i=1}^m) = k] \\
&= \sum_{k=1}^{n-m+1} \frac{k\binom{n-k}{m-1}}{\binom{n}{m}} \\
&= \frac{1}{\binom{n}{m}} \sum_{k=1}^{n-m+1} k\binom{n-k}{m-1} \\
&= \frac{1}{\binom{n}{m}} \sum_{k=1}^{n-m+1} \sum_{i=k}^{n-m+1} \binom{n-i}{m-1} \qquad \text{(rearranging)} \\
&= \frac{1}{\binom{n}{m}} \sum_{k=1}^{n-m+1} \binom{n-k+1}{m} \qquad \text{(using the hockey-stick identity)} \\
&= \frac{\binom{n+1}{m+1}}{\binom{n}{m}} \qquad \text{(using the hockey-stick identity)} \\
&= \frac{\frac{(n+1)!}{(n-m)!(m+1)!}}{\frac{n!}{(n-m)!m!}} \\
&= \frac{n+1}{m+1},
\end{aligned} \tag{8}
$$

which concludes the proof. □

**Proof of Theorem 1** For $\rho_{\mathrm{T}}(\mathcal{M}, f) \neq 0$, it can be regarded as taking $\mathbb{A}_{T,\mathcal{M},f}$ where $|\mathbb{A}_{T,\mathcal{M},f}| = \rho_{\mathrm{T}}(\mathcal{M}, f)T$ out of the top $T$ architectures in $f$. As $R_f(\mathcal{A}^*_{\mathcal{M},T}) = \min(R_f(\mathcal{A}), \mathcal{A} \in \mathbb{A}_{T,\mathcal{M},f})$. Combining with the uniform distribution assumption $\mathbb{P}[R_f(A) = t] = 1/T$, the result can be directly derived from Lemma 1, which concludes the proof.

**Remark** The uniform distribution assumption $\mathbb{P}[R_f(A) = t] = 1/T$ is based on the fact, given that we do not have prior knowledge about the distribution of the estimation metric $M$ and the objective evaluation metric $f$, we assume any permutation of architectures has equal probability to be produced by $M$ and $f$, i.e., probability of $1/N!$, where $N$ is the number of architectures. This is aligned with the expectation computation of uniform randomness as stated in Lemma 1.

### A.3 Proof of Theorem 2

Before we present the proof, let us first formally introduce the definition of linear partial monitoring game, global observability, information directed sampling (IDS), and their extension to reproducing kernel Hilbert Spaces (RKHS) (i.e., BO). These concepts are proposed by (Kirschner et al., 2020). Additionally, we will introduce an important theorem that establishes the regret bound for a globally observable game. Subsequently, we will demonstrate how our algorithm fits into this framework and derive the upper bound on the expected ranking of $\widetilde{\mathcal{A}}^*_{\mathbb{M},T}$. Some proof tricks are inspired from (Chaudhuri & Tewari, 2017).

**Linear Partial Monitoring Game** Let $\mathcal{X} \subset \mathbb{R}^d$ be an action set, and $\theta \in \mathbb{R}^d$ be the unknown parameter to generate the reward. For an action $x \in \mathcal{X}$, the reward is given as $\langle x, \theta \rangle$, and there is a corresponding **known** linear observation operator $A_x \in \mathbb{R}^{d \times m}$ that produces an $m$-dimension observation as $a = A_x^\top \theta + \epsilon$ where $\epsilon$ is an $\xi$-subgaussian noise vector. Suppose the game is played for $n$ rounds; in round $t$, the learner selects action $x_t$. The goal of the game is to minimize the cumulative regret $Reg_n = \sum_{t=1}^n \langle x^* - x_t, \theta \rangle$, where $x^* = \arg\max_{x \in \mathcal{X}} \langle x, \theta \rangle$ represents the optimal action.

**Global Observability** A linear partial monitoring game is global observable iff $\forall x, y \in \mathcal{X}, x-y \in \text{span}(A_z, z \in \mathcal{X})$. Here, $\text{span}(A_z, z \in \mathcal{X})$ is defined as the span of all the columns of the matrices $(A_z, z \in \mathcal{X})$.

**IDS** IDS is a sampling strategy to minimize the information ratio when sampling a new action $x$ from the proposed distribution $\mu$. Specifically, in the round $t$, the proposed distribution is defined as $\mu_t = \arg\min_\mu \frac{\mathbb{E}_\mu[\Delta_t(x)]^2}{\mathbb{E}_\mu[I_t(x)]}$, where $\Delta_t(x)$ is the conservative gap estimate such that $\Delta_t(x) \geq \langle x^* - x, \theta \rangle$ and $I_t(x)$ is the information gain.

**Partial Monitoring in RKHS** Also known as the kernelized setting of partial monitoring, the action set now is defined as $\mathcal{X}_0$, which is not necessarily a subset of $\mathbb{R}^d$. For every action $x \in \mathcal{X}_0$, it non-linearly depends on the features through a positive-definite kernel $k : \mathcal{X}_0 \times \mathcal{X}_0 \to \mathbb{R}$. Let $\mathcal{H}$ be the corresponding RKHS of the given kernel $k$. The unknown parameter is now defined as $f \in \mathcal{H}$ (instead of $\theta$), and for the given action $x$, the reward takes the form of $f(x) = \langle k_x, f \rangle$. The known observation operator is now defined as $A_x : \mathcal{H} \to \mathbb{R}^m$, and thus the observation is $a = A_x f + \epsilon$. The cumulative regret for a game played for $n$ rounds is $Reg_n = \sum_{t=1}^n f(x^*) - f(x) = \langle k_{x^*} - k_x, f \rangle$ where $x^* = \arg\max_{x \in \mathcal{X}_0} f(x)$. The game is global observable iff $\forall x, y \in \mathcal{X}_0, k_x - k_y \in \text{span}(A_z, z \in \mathcal{X}_0)$.

**Theorem 3** (Corollary 18 in (Kirschner et al., 2020))**.** *For the kernelized setting of partial monitoring using IDS as a sampling policy, if the game is global observable, then* $Reg_n \leq \mathcal{O}(n^{2/3}(\alpha\beta_n(\gamma_n + \log\frac{1}{\delta})^{1/3})$ *with probability at least* $1 - \delta$ *where* $\alpha$ *is the alignment constant where the value is bounded for the global observable game, and* $\beta_n$*,* $\gamma_n$ *are variables related to* $n$ *but only logarithmically depend on* $n$*, i.e.,* $\beta_n, \gamma_n \leq \mathcal{O}(\log n)$*.*

Next, we would like to elaborate on the two conditions we assume to be satisfied.

**Expressiveness of the Hypothesis Space** For any two weight vectors $\boldsymbol{w}_1, \boldsymbol{w}_2$, let $\mathbb{A}_{T,\mathcal{M}(\cdot;\boldsymbol{w}_1)} = \{\mathcal{A}, R_{\mathcal{M}(\cdot;\boldsymbol{w}_1)}(\mathcal{A}) \leq T\}$ and $\mathbb{A}_{T,\mathcal{M}(\cdot;\boldsymbol{w}_2)} = \{\mathcal{A}, R_{\mathcal{M}(\cdot;\boldsymbol{w}_2)}(\mathcal{A}) \leq T\}$. Then $\forall \mathcal{A} \in \mathbb{A}_{T,\mathcal{M}(\cdot;\boldsymbol{w}_1)} \Delta \mathbb{A}_{T,\mathcal{M}(\cdot;\boldsymbol{w}_2)}$, $\exists \boldsymbol{w'}, R_{\mathcal{M}(\cdot;\boldsymbol{w'})}(\mathcal{A}) = 1$. In simpler terms, this condition necessitates that if the top $T$ architectures of the estimation metrics generated by two weight vectors differ, there must always be a way to evaluate the performance of these architectures through a linear combination. Consequently, the hypothesis space produced by the linear combination must be sufficiently expressive. In practice, to enhance the expressiveness of the linear combination, we normalize the values of training-free metrics to the range $[0, 1]$, before performing the linear combination. This normalization increases the likelihood that more architectures can be ranked first by some weight vectors, as no single training-free metric dominates the estimation metric. Our experiments and ablation studies suggest that this condition generally holds for the training-free metrics proposed in the literature and the benchmarks commonly employed in the community.

**Predictable Ranking through Performance** For $\forall \mathcal{A}$, suppose $f(\mathcal{A})$ is known, then $\exists \widetilde{R}_f, \mathbf{1}\{\widetilde{R}_f(\mathcal{A}) \leq T\} = \mathbf{1}\{R_f(\mathcal{A}) \leq T\} + \epsilon$ where $\epsilon$ is an $\xi$-subgaussian noise. This assumption requires that we can approximately determine if an architecture is top $T$ in $f$ given its performance, which means the ranking is predictable by the performance. Notice that this ranking threshold estimator does not need to be explicitly specified, but is used to ensure that for our Algorithm 1, observing $f(\mathcal{A})$ is approximately equivalent to observing $\mathbf{1}\{R_f(\mathcal{A}) \leq T\}$ with search budget $T$. In practice, the benchmark widely used in the community generally satisfies such predictable requirements.

To begin with our proof for Theorem 2, we first need to prove the global observability of the game to evaluate the expectation of $\rho_{\text{T}}(\mathcal{M}(\cdot; \widetilde{\boldsymbol{w}}^*), f)$.

**Theorem 4.** *If the conditions of* ***Expressiveness of the Hypothesis Space*** *and* ***Predictable Ranking through Performance*** *can be satisfied, then*

$$\mathbb{E}[\rho_T(\mathcal{M}(\cdot; \widetilde{\boldsymbol{w}}^*), f)] \geq \rho_T^*(\mathcal{M}_{\mathbb{M}}, f) - q_T T^{-1/3},$$

*with probability arbitrary close to 1 where* $\rho_T^*(\mathcal{M}_{\mathbb{M}}, f)$ *is defined in Theorem 2,* $q_T = C\alpha\beta_T\gamma_T^{1/3}$ *where* $\alpha, \beta_T, \gamma_T$ *is specified in Theorem 3, and* $C > 0$ *is a universal constant.*

*Proof.* To fit our algorithm into the framework of the kernelized setting of partial monitoring, we set the action as the weight vector $\boldsymbol{w}$ and the corresponding reward as $\rho_{\mathrm{T}}(\mathcal{M}(\cdot;\boldsymbol{w}), f)$. To derive the closed-form of the kernel $k_{\boldsymbol{w}}$, the observation operator $A_{\boldsymbol{w}}$ and the unknown parameter $\theta$ (to prevent ambiguity with the objective evaluation metric $f$, we still denote the unknown parameter generating the reward $\langle k_{\boldsymbol{w}}, \theta\rangle$ as $\theta$), we propose to configure $k_{\boldsymbol{w}}$, $A_{\boldsymbol{w}}$ and $\theta$ as $N$-dimension vectors where $N$ is the number of the architectures, and $(k_{\boldsymbol{w}})_i = \mathbf{1}(R_{\mathcal{M}(\cdot;\boldsymbol{w})}(\mathcal{A}_i) \leq T)$, $(A_{\boldsymbol{w}})_i = \mathbf{1}(R_{\mathcal{M}(\cdot;\boldsymbol{w})}(\mathcal{A}_i) = 1)T$, $(\theta)_i = \mathbf{1}(R_f(\mathcal{A}_i) \leq T)/T$.

Intuitively, $(k_{\boldsymbol{w}})_i$ denotes whether $\mathcal{A}_i$ is within the top $T$ architectures under $\mathcal{M}(\cdot;\boldsymbol{w})$, $(A_{\boldsymbol{w}})_i$ indicates whether $\mathcal{A}_i$ is the top architecture under $\mathcal{M}(\cdot;\boldsymbol{w})$, and $(\theta)_i$ reveals whether $\mathcal{A}_i$ is among top $T$ architectures under $f$. Therefore, the observation is given as $a_{\boldsymbol{w}} = A_{\boldsymbol{w}}^{\top}\theta + \epsilon$, where $a_{\boldsymbol{w}}$ equals to $\mathbf{1}(R_f(\mathcal{A}(\boldsymbol{w})) \leq T) + \epsilon$, indicating whether $\mathcal{A}(\boldsymbol{w})$ is a top $T$ architecture under $f$. Here, $\mathcal{A}(\boldsymbol{w})$ represents the top architecture under $\mathcal{M}(\cdot;\boldsymbol{w})$, as per the earlier notation. With a ranking threshold predictor $\widetilde{R}_f$ as discussed in **Predicatable Ranking through Performance** and the performance of $f(\mathcal{A}(\boldsymbol{w}))$, we can provide such observation with a $\xi$-subgaussian noise $\epsilon$. The reward is computed as $\langle k_{\boldsymbol{w}}, \theta\rangle$, which corresponds to $\rho_{\mathrm{T}}(\mathcal{M}(\cdot;\boldsymbol{w}), f)$. Notice that for any given weight vector $\boldsymbol{w}$, the values of $k_{\boldsymbol{w}}$ and $A_{\boldsymbol{w}}$ are always known to the learner (i.e., the learner always which architecture(s) are the top $T$/top 1 architecture(s) under $\mathcal{M}(\cdot;\boldsymbol{w})$), thereby fitting our algorithm within the kernelized setting of partial monitoring.

When the condition of **Expressiveness of the Hypothesis Space** is satisfied, the condition can be directly used to derive the global observability, as $k_{\boldsymbol{w}_1} - k_{\boldsymbol{w}_2}$ literally refers to the set difference $\mathbb{A}_{T,\mathcal{M}(\cdot;\boldsymbol{w}_1)} \Delta \mathbb{A}_{T,\mathcal{M}(\cdot;\boldsymbol{w}_2)}$.

With the Theorem 3, we can obtain

$$\begin{aligned}\mathbb{E}[\rho_{\mathrm{T}}(\mathcal{M}(\cdot;\widetilde{\boldsymbol{w}}^*), f)] &= \rho_{\mathrm{T}}^*(\mathcal{M}_{\mathbb{M}}, f) - \frac{Reg_T}{T} \\ &\geq \rho_{\mathrm{T}}^*(\mathcal{M}_{\mathbb{M}}, f) - \frac{q_T T^{2/3}}{T} \\ &= \rho_{\mathrm{T}}^*(\mathcal{M}_{\mathbb{M}}, f) - q_T T^{-1/3},\end{aligned}$$

which concludes the proof. □

**Proof of Theorem 2** As we have obtained the lower bound of $\mathbb{E}[\rho_{\mathrm{T}}(\mathcal{M}(\cdot;\widetilde{\boldsymbol{w}}^*), f)]$, and as we search for $T - T_0$ architectures for exploitation to obtain $\widetilde{\mathcal{A}}^*_{\mathbb{M},T}$ as stated in Section 4.3, Theorem 2 can be directly derived from Theorem 1 and Lemma 1, which concludes the proof.

### A.4 Derivation for the claim about the Influence of T in Section 5.2

Suppose that $\rho_{\mathrm{T}}^*(\mathcal{M}, f) = \max_{\mathcal{M}\in\mathbb{M}}(\rho_{\mathrm{T}}(\mathcal{M}, f))$, then $\mathbb{E}[R_f(\widetilde{\mathcal{A}}^*_{\mathbb{M},T})] \leq \min_{\mathcal{M}\in\mathbb{M}}(\mathbb{E}[R_f(\mathcal{A}^*_{\mathcal{M},T})])$ implies $\epsilon \geq q_T T^{-1/3}/\rho_{\mathrm{T}}^*(\mathcal{M}, f) + 1/(1-\alpha)$, where $\alpha = T_0/T$ and $\epsilon = \rho_{\mathrm{T}}^*(\mathcal{M}_{\mathbb{M}}, f)/\rho_{\mathrm{T}}^*(\mathcal{M}, f)$. The right-hand side can be regarded as the threshold to define how good $\rho_{\mathrm{T}}^*(\mathcal{M}_{\mathbb{M}}, f)$ should be so that the outperforming can be achieved, and this threshold decreases when $T$ increases, which supports our initial claim.

## Appendix B Experimental Details

### B.1 Benchmark Information

**NAS-Bench-201 (Dong & Yang, 2020)** NAS-Bench-201 is a widely used NAS benchmark that has served as the testing ground for various training-based NAS algorithms (Pham et al., 2018; Liu et al., 2019; Dong & Yang, 2019) as well as training-free NAS metrics (Abdelfattah et al., 2020; Mellor et al., 2021). The main focus of its search space is the structure of a cell in a neural architecture. In this context, a cell is composed of 4 nodes interconnected by 6 edges. Each edge can be associated with one of five operations: $3\times3$ convolution, $1\times1$ convolution, $3\times3$ average pooling, zeroize, or skip connection. Consequently, the search space consists of $5^6 = 15{,}625$ unique neural architectures. These architectures are evaluated across three datasets: CIFAR-10 (C10) (Krizhevsky et al., 2009), CIFAR-100 (C100), and ImageNet-16-120 (IN-16) (Chrabaszcz et al., 2017).

**TransNAS-Bench-101 (Duan et al., 2021)** TransNAS-Bench-101 is a recent addition to the NAS benchmarks and has been less explored by NAS algorithms. Unlike NAS-Bench-201, which focuses solely on the cell structure, TransNAS-Bench-101 explores both micro-cell structure and macro skeleton structure, leading to two distinct search spaces: micro and macro. The micro search space resembles NAS-Bench-201's structure but features only four operations (omitting the average pooling operation), resulting in a total of $4^6 = 4{,}096$ neural architectures. In the macro search space, the cell structure is fixed while the skeleton is varied. The skeleton contains 4 to 6 modules, each with two residual blocks. Each module can opt to downsample the feature map, double the channels, or do both. Across the entire skeleton, downsampling can occur 1 to 4 times, and channel doubling can happen 1 to 3 times, yielding a total of 3,256 architectures. TransNAS-Bench-101 evaluates these architectures on seven different vision tasks, all using a single dataset of 120K indoor scene images, derived from the Taskonomy project (Zamir et al., 2018). The tasks include scene classification (Scene), object detection (Object), jigsaw puzzle (Jigsaw), room layout (Layout), semantic segmentation (Segment.), surface normal (Normal), and autoencoding (Autoenco.).

**DARTS (Liu et al., 2019)** This search space searches on two cells, where each cell has 2 input nodes and 4 intermediate nodes with 2 predecessors each. On each edge between two nodes, it can have 7 non-zero operation choices. DARTS search space is not a tabular benchmark, as it contains a total of $10^{18}$ unique architectures. It evaluates architectures in 3 datasets: CIFAR-10 (C10), CIFAR-100 (C100), and ImageNet (Deng et al., 2009) datasets.

### B.2 Implementation Details of RoBoT

**Algorithm 1 Details** In Section 4.2, we briefly discussed the use of *Bayesian optimization*(BO) in our algorithm. We would now delve into the implementation of the Gaussian Process (GP) to construct the prior and posterior distributions and the choice of our acquisition function for determining the next queried weight vector. Given a set of observations $[f(\mathcal{A}(\boldsymbol{w}_1)), \ldots, f(\mathcal{A}(\boldsymbol{w}_t))]$, we assume that they are randomly drawn from a prior probability distribution, in this case, a GP. The GP is defined by a mean function covariance (or kernel) function. We set the mean function to be a constant, such as 0, and choose the Matérn kernel for the kernel function. Based on these observations and the prior mean and kernel functions, we calculate the posterior mean and kernel function as $\mu(\boldsymbol{w})$ and $k(\boldsymbol{w}, \boldsymbol{w}')$, respectively, following Equation (1) in (Srinivas et al., 2010). We then derive the variance as $\sigma^2(\boldsymbol{w}) = k(\boldsymbol{w}, \boldsymbol{w})$. As for the acquisition function, we adopt the upper confidence bound (UCB) (Srinivas et al., 2010), as suggested by Kirschner et al. (2020) for its deterministic IDS properties. The next weight vector is chosen as $\boldsymbol{w}_{t+1} = \arg\max_{\boldsymbol{w}} \mu(\boldsymbol{w}) + \kappa\sigma(\boldsymbol{w})$, where $\kappa$ is the exploration-exploitation trade-off constant that regulates the balance between exploring the weight vector space and exploiting the current regression results.

Owing to the capabilities of BO in solving black-box problems through global optimization, coupled with its efficiency, it has enjoyed a prolonged period of application within the field of NAS (Cao et al., 2018; White et al., 2021a; Shu et al., 2022b; Shen et al., 2023). Although there is a substantial body of theoretical research in the field of BO (Srinivas et al., 2010; Dai et al., 2019; 2020; 2022), and considerable exploration of theoretical studies within the realm of NAS (Ning et al., 2021; Shu et al., 2019; 2022b), to the best of our knowledge, this work represents the first instance of applying theoretical findings from the domain of BO to the field of NAS, thereby proposing a bounded expected performance with theoretical backing. For a more comprehensive understanding of BO, we refer to (Srinivas et al., 2010), and for implementation details, we refer to (Nogueira, 2014–), both of which guided our experimental setup.

**Implementation Details of RoBoT on NAS-Bench-201 and TransNAS-Bench-101** For both NAS-Bench-201 and TransNAS-Bench-101, we utilize the six training-free metrics outlined in (Abdelfattah et al., 2020): *grad_norm*, *snip*, *grasp*, *fisher*, *synflow*, and *jacob_cov*. We pre-calculate these metrics' values for all neural architectures across all tasks. To ensure reproducibility, we directly utilize the computed training-free metrics from Zero-Cost-NAS (Abdelfattah et al., 2020) and NAS-Bench-Suite-Zero (Krishnakumar et al., 2022) for NAS-Bench-201 and TransNAS-Bench-101, respectively. For a particular task, we normalize a metric's values to fit within the [0, 1] range. We define the search range of $\boldsymbol{w}$ to be any real value between -1 and 1, given the consideration that it's used for a weighted linear combination to rank architectures. The ranking depends on relative weights rather than their absolute magnitudes, hence a normalized range covers all real space

equivalently for the purpose of ranking. For NAS-Bench-201, we use the CIFAR-10 validation performance after 12 training epochs (i.e., "hp=12") from the tabular data in NAS-Bench-201 as the objective evaluation metric $f$ for all three datasets and compute the search cost displayed in Table 2 in the same manner (which is the training cost of 20 architectures). However, we report the full training test accuracy of the proposed architecture after 200 epochs. As for TransNAS-Bench-101, we note that for tasks *Segment.*, *Normal*, and *Autoenco.* on both micro and macro datasets, the training-free metric *synflow* is inapplicable due to a tanh activation at the architecture's end, so we only use the remaining five training-free metrics. Moreover, given the considerable gap between validation and test performances in TransNAS-Bench-101, we only report our proposed architecture's validation performance.

As for the DARTS search space, we refer to Section C.2 for the implementation details.

**Reported Search Costs of RoBoT** For Table 2, Table 4 and Table 7, the search costs of RoBoT consist of three components: the computation costs of training-free metrics of the entire search space (which we treat as negligible in early discussion), the queries for objective evaluation performance in BO, and in greedy search. To reduce the search costs incurred in queries for objective evaluation performance, we follow (Shu et al., 2022b) to apply the validation performance of each candidate architecture that is trained from scratch for only a limited number of epochs (e.g., 12 epochs in NAS-Bench-201) to approximate the true architecture performance, which in fact is quite computationally efficient (e.g., about 110 GPU-seconds for the model training of each candidate in NAS-Bench-201, and 0.04 GPU-day for ImageNet in DARTS search space). Of note, such an approximation is already reasonably good to help find well-performing architectures, as supported by our results.

### B.3 Implementation Details of NAS Baselines

Throughout our experiments, we focus on the following query-based NAS algorithms:

**Random Search (RS)** As implied by its name, RS randomly assembles a pool of architectures, evaluates the performance of each, and proposes the highest-performing one. Despite its simplicity, this method often proves a strong baseline for NAS algorithms (Yu et al., 2019; Li & Talwalkar, 2020), matching the performance stated in Lemma 1 .

**Regularized Evolutionary Algorithm (REA) (Real et al., 2019)** Like RS, REA initiates by assembling a small pool of architectures and evaluating each one's performance. In our experiments, the initial pool size is chosen to be a third of the total search budget, $T/3$. Then, the top-performing architecture is removed from the pool and applied to a *mutation* to create a new architecture. This new architecture is evaluated and added to the pool. In our context, a mutation involves altering one operation on an edge within the cell (or for TransNAS-Bench-101-macro, we randomly add or remove a downsample/doubling operation on one module, and ensure the generated architecture is valid). As it is typically assumed that a good-performing architecture's neighbor will perform better than a randomly chosen architecture, REA is expected to outperform RS.

**REINFORCE (Williams, 1992)** REINFORCE is a reinforcement learning algorithm that utilizes the objective evaluation metric as the reward to update the policy of choosing architectures. We follow the configuration outlined in (Dong & Yang, 2020) to use Adam (Kingma & Ba, 2015) with the learning rate of 0.01 as the optimizer to update the policy and use the exponential moving average with the momentum of 0.9 as the reward baseline.

**Hybrid Neural Architecture Search (HNAS) (Shu et al., 2022b)** Shu et al. (2022b) aims to select the architecture with the minimal upper bound on generalization error, where the upper bound for each architecture is specified with the corresponding training-free metric value and two unknown hyperparameters. HNAS employs BO with the observation of the objective evaluation metric performance of the chosen architecture to optimize the values of two hyperparameters. As HNAS requires that the training-free metric should be gradient-based, we follow their recommendation and use *grad_norm* as the training-free metric for our experiments.

**Average Training-Free Metric Performance (Avg.)** We apply greedy search as stated in Section 4.3 to traverse the top $T$ architectures for each training-free metric and select the architecture

Table 5: Comparison of NAS algorithms in NAS-Bench-201 regarding the ranking of test performance.All methods query for the validation performance for 20 models.

| **Algorithm** | **Test Ranking** | | |
|---|---|---|---|
| | C10 | C100 | IN-16 |
| REA | 814±1175 | 789±891 | 938±1170 |
| RS (w/o sharing) | 1055±981 | 901±900 | 815±730 |
| REINFORCE | 995±1042 | 899±814 | 722±596 |
| HNAS | 212±310 | 202±257 | 183±151 |
| Training-Free | | | |
| ↪ Avg. | 3331±2926 | 4514±3082 | 5256±3785 |
| ↪ Best | **3** | **1** | **24** |
| RoBoT | **3**±0 | **1**±0 | **24**±0 |

with the highest objective evaluation performance. The reported performance is the average across all these selected architectures for each training-free metric.

**Best Training-Free Metric Performance (Best)** The experimental setup is identical to that of Avg., however, in this case, we report the performance of the single best architecture across all those selected for each training-free metric.

# APPENDIX C MORE EMPIRICAL RESULTS

## C.1 ADDITIONAL RESULTS ON NAS BENCHMARK

In this section, we report additional empirical results on NAS-Bench-201 and TransNAS-Bench-101. The results are provided in Table 5, 6, and Figure 3, 9, 10. All the figures on the left side demonstrate the objective evaluation performance while the figures on the right side demonstrate the corresponding ranking. These additional results align well with our initial results and arguments presented in Section 6.1, further validating that RoBoT offers reliable and top-tier performance across various tasks, achieving competitive as the best training-free metric's performance without prior knowledge about which training-free metric will perform the best. Particularly, RoBoT exhibits a clear superior performance in tasks like *micro-Object*, *micro-Jigsaw*, *micro-Normal*, *macro-Object*, and *macro-Jigsaw*, demonstrating its potential to boost training-free metrics beyond their original performance. Compared with HNAS, our proposed RoBoT shows more stable and higher performance. Despite the decent performance of HNAS in a few tasks such as *micro-Segment.*, it struggles with inconsistency issues inherent to single training-free metrics, leading to overall worse performance. In contrast, RoBoT consistently delivers robust and competitive performance, further emphasizing its value for NAS. These additional results strengthen our initial conclusions, confirming the robustness, consistency, and superior performance of RoBoT across various tasks on both NAS-Bench-201 and TransNAS-Bench-101 benchmarks.

## C.2 ROBOT IN THE DARTS SEARCH SPACE

To further verify the robustness of our algorithm RoBoT, we also implement RoBoT in the DARTS (Liu et al., 2019) search space, aiming to identify high-performing architectures in the CIFAR-10/100 and ImageNet (Deng et al., 2009) datasets. We create a pool of 60000 architectures and evaluate their training-free metrics (same as NAS-Bench-201 and Trans-NAS-Bench-101, we evaluate six training-free metrics: *grad_norm*, *snip*, *grasp*, *fisher*, *synflow*, and *jacob_cov*, and normalize each metric's values to fit within the [0, 1] range) on the corresponding datasets. With regards to CIFAR-10/100, RoBoT is given a search budget $T = 25$, to query for the performance of the selected architectures with a 10-epoch model training. As for ImageNet, we use a search budget $T = 10$, and the performance of the selected architectures is determined when they are trained for 3 epochs. As per the methodology in the DARTS study (Liu et al., 2019), we built 20-layer final selected architectures. These architectures have 36 initial channels and an auxiliary tower with a weight of 0.4 for

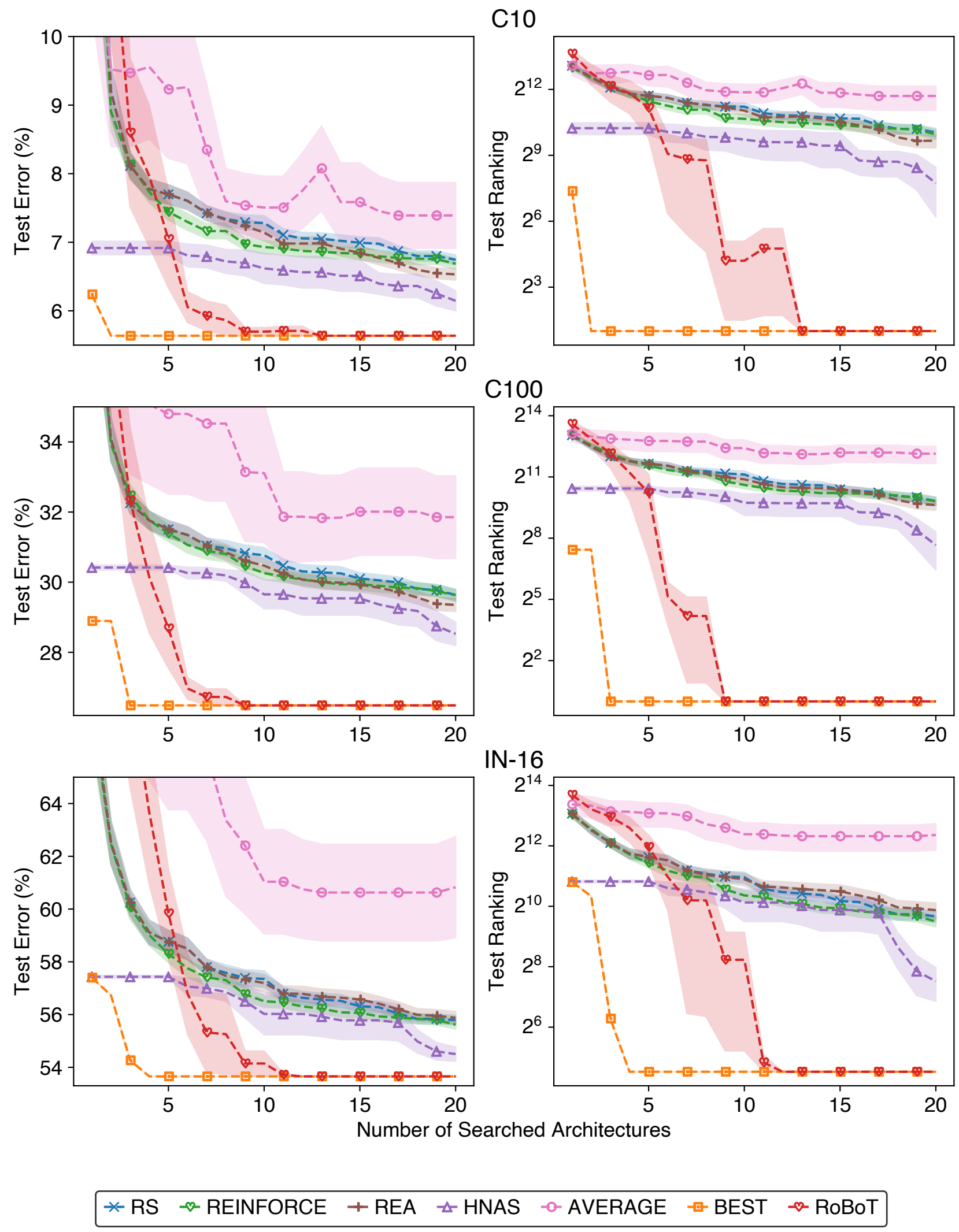

Figure 3: Comparison between RoBoT and other NAS baselines in NAS-Bench-201 regarding the number of searched architectures. Note that RoBoT and HNAS are reported with the mean and standard error of 10 independent searches, while RS, REA, and REINFORCE are reported with 50 independent searches.

CIFAR-10 and 0.6 for CIFAR-100, located at the 13th layer. We test these architectures on CIFAR-10/100 by employing stochastic gradient descent (SGD) over 600 epochs. The learning rate started at 0.025 and gradually reduced to 0 for CIFAR-10, and from 0.035 to 0.001 for CIFAR-100, using a cosine schedule. The momentum was set at 0.9 and the weight decay was $3\times10^{-4}$ with a batch size of 96. Additionally, we use Cutout (Devries & Taylor, 2017) and ScheduledDropPath, which linearly increased from 0 to 0.2 for CIFAR-10 (and from 0 to 0.3 for CIFAR-100) as regularization techniques for CIFAR-10/100. For the ImageNet evaluation, we train a 14-layer architecture from scratch over 250 epochs, with a batch size of 1024. The learning rate was initially increased to 0.7 over the first 5 epochs and then gradually decreased to zero following a cosine schedule. The SGD optimizer was used with a momentum of 0.9 and a weight decay of $3\times10^{-5}$.

We present the results in Table 7 and Table 4. Notably, despite lacking prior information on the performance of these training-free metrics in the DARTS search space, our proposed algorithm, RoBoT, still delivers competitive results when compared with other NAS techniques. It is worth noting the significant discrepancy between the validation performance (i.e., the performance of the architecture when trained for 10 epochs/3 epochs on CIFAR-10/100 / ImageNet, respectively) and the final test performance (when trained for 600 epochs/250 epochs on CIFAR-10/100 / ImageNet, respectively).

Table 6: Comparison of NAS algorithms in TransNAS-Bench-101-micro regarding validation performance. The results of RoBoT and HNAS are reported with the mean and the standard deviation of 10 independent searches, while 50 independent searches for REA, RS, and REINFORCE.

| **Space** | **Algorithm** | Accuracy (%) | | | L2 Loss ($\times 10^{-2}$) | mIoU (%) | SSIM ($\times 10^{-2}$) | |
|---|---|---|---|---|---|---|---|---|
| | | Scene | Object | Jigsaw | Layout | Segment. | Normal | Autoenco. |
| **Micro** | REA | 54.63±0.18 | 44.92±0.38 | 94.81±0.21 | −62.06±0.69 | 94.55±0.03 | 57.10±0.51 | **56.23**±0.81 |
| | RS (w/o sharing) | 54.56±0.22 | 44.76±0.39 | 94.63±0.26 | −62.22±0.96 | 94.53±0.03 | 56.83±0.46 | 55.55±0.99 |
| | REINFORCE | 54.56±0.19 | 44.69±0.34 | 94.70±0.23 | −62.20±0.83 | 94.53±0.03 | 56.96±0.43 | 55.75±0.86 |
| | HNAS | 54.29±0.09 | 44.08±0.00 | 94.56±0.21 | −64.83±1.69 | 94.57±0.00 | 56.88±0.00 | 48.66±0.00 |
| | Training-Free | | | | | | | |
| | ↪ Avg. | 54.60±0.36 | 43.98±0.87 | 94.11±0.54 | −64.27±1.27 | 94.03±1.07 | 52.26±9.33 | 41.36±17.29 |
| | ↪ Best | **54.87** | **45.59** | 94.76 | −62.12 | **94.58** | 57.05 | 55.30 |
| | RoBoT | **54.87**±0.00 | **45.59**±0.00 | **94.82**±0.06 | **−61.16**±0.86 | **94.58**±0.00 | **57.44**±0.34 | 55.42±1.05 |
| | **Optimal** | 54.94 | 45.59 | 95.37 | −60.10 | 94.61 | 58.73 | 57.72 |
| **Macro** | REA | 56.69±0.34 | 46.74±0.33 | 96.78±0.10 | −59.99±1.09 | 94.80±0.03 | 60.81±0.72 | 71.38±2.49 |
| | RS (w/o sharing) | 56.53±0.28 | 46.68±0.30 | 96.74±0.19 | −60.27±1.08 | 94.78±0.04 | 60.72±0.72 | 70.05±3.01 |
| | REINFORCE | 56.43±0.29 | 46.67±0.29 | 96.80±0.14 | −60.29±1.00 | 94.78±0.04 | 60.48±0.94 | 69.21±2.55 |
| | HNAS | 55.03±0.00 | 45.00±0.00 | 96.28±0.18 | −61.40 ±0.11 | 94.79±0.00 | 59.27±0.00 | 57.59±0.00 |
| | Training-Free | | | | | | | |
| | ↪ Avg. | 55.81±1.03 | 45.87±0.87 | 96.44±0.31 | −61.10±1.20 | 94.78±0.04 | 61.19±0.39 | 70.93±2.84 |
| | ↪ Best | **57.41** | 46.87 | 96.89 | **−58.44** | 94.83 | **61.66** | 73.51 |
| | RoBoT | 57.35±0.13 | **46.94**±0.09 | **96.92**±0.02 | −58.88±0.70 | **94.85**±0.02 | **61.66**±0.00 | **73.53**±0.06 |
| | **Optimal** | 57.41 | 47.42 | 97.02 | −58.22 | 94.86 | 64.35 | 74.88 |

Table 7: Performance comparison among state-of-the-art (SOTA) neural architectures on CIFAR-10/100. The performance of the final architectures selected by RoBoT is reported with the mean and standard deviation of five independent evaluations. The search costs are evaluated on a single Nvidia 1080Ti.

| **Algorithm** | **Test Error (%)** | | **Params** (M) | | **Search Cost** (GPU Hours) | **Search Method** |
|---|---|---|---|---|---|---|
| | C10 | C100 | C10 | C100 | | |
| DenseNet-BC (Huang et al., 2017) | 3.46* | 17.18* | 25.6 | 25.6 | - | manual |
| NASNet-A (Zoph et al., 2018) | 2.65 | - | 3.3 | - | 48000 | RL |
| AmoebaNet-A (Real et al., 2019) | 3.34±0.06 | 18.93† | 3.2 | 3.1 | 75600 | evolution |
| PNAS (Liu et al., 2018) | 3.41±0.09 | 19.53* | 3.2 | 3.2 | 5400 | SMBO |
| ENAS (Pham et al., 2018) | 2.89 | 19.43* | 4.6 | 4.6 | 12 | RL |
| NAONet (Luo et al., 2018) | 3.53 | - | 3.1 | - | 9.6 | NAO |
| DARTS (2nd) (Liu et al., 2019) | 2.76±0.09 | 17.54† | 3.3 | 3.4 | 24 | gradient |
| GDAS (Dong & Yang, 2019) | 2.93 | 18.38 | 3.4 | 3.4 | 7.2 | gradient |
| NASP (Yao et al., 2020) | 2.83±0.09 | - | 3.3 | - | 2.4 | gradient |
| P-DARTS (Chen et al., 2019) | 2.50 | - | 3.4 | - | 7.2 | gradient |
| DARTS- (avg) (Chu et al., 2020) | 2.59±0.08 | 17.51±0.25 | 3.5 | 3.3 | 9.6 | gradient |
| SDARTS-ADV (Chen & Hsieh, 2020) | 2.61±0.02 | - | 3.3 | - | 31.2 | gradient |
| R-DARTS (L2) (Zela et al., 2020) | 2.95±0.21 | 18.01±0.26 | - | - | 38.4 | gradient |
| DrNAS (Chen et al., 2021b) | 2.46±0.03 | - | 4.1 | - | 14.4 | gradient |
| TE-NAS (Chen et al., 2021a) | 2.83±0.06 | 17.42±0.56 | 3.8 | 3.9 | 1.2 | training-free |
| NASI-ADA Shu et al. (2021) | 2.90±0.13 | 16.84±0.40 | 3.7 | 3.8 | 0.24 | training-free |
| HNAS (Shu et al., 2022b) | 2.62±0.04 | 16.29±0.14 | 3.4 | 3.8 | 2.6 | hybrid |
| RoBoT | 2.60±0.03 | 16.52±0.10 | 3.3 | 3.8 | 3.5 | hybrid |

While our algorithm RoBoT is not explicitly designed to address this gap, it still demonstrates its effectiveness by identifying high-performing architectures. Overall, these results further substantiate our previous assertions about the robustness of our algorithm RoBoT in Section 6, suggesting that our approach can be effectively applied in practical, real-world scenarios.

### C.3 More Ablation Studies

To delve deeper into the factors impacting RoBoT and to provide interesting insights, we carry out several ablation studies as detailed below. These studies focus on the tasks of *Scene*, *Object*, *Jigsaw*, *Segement.*, *Normal* on TransNAS-Bench-101-micro. The ablation studies explore various aspects of the algorithm and provide valuable findings specific to these tasks.

**Ablation Study on Optimized Linear Combination Weights** To understand the influence of the optimized linear combination weights $\widetilde{\boldsymbol{w}}^*$ that used to formulate $\mathcal{M}(\cdot; \widetilde{\boldsymbol{w}}^*)$, we present the varying weights on the tasks of *Scene*, *Object*, *Jigsaw*, *Layout* in TransNAS-Bench-101-micro as well as their similarity and correlation in Table 8 and Figure 4. The results show that the optimized weights typically vary for different tasks, which aligns with the observations and motivations in our Section 3 and further highlights the necessity of developing robust metrics that can perform consistently well on diverse tasks such as our RoBoT. In addition, for tasks with similar characteristics, e.g., the *Scene* and *Object* tasks, both of which are classification tasks, the optimized weights share a relatively high similarity/correlation, indicating the potential transferability of the optimized linear combination within similar tasks.

Table 8: The varying optimized linear combination weights on 4 tasks of TransNAS-Bench-101-micro.

| **Tasks** | grad_norm | snip | grasp | fisher | synflow | jacob_cov |
|---|---|---|---|---|---|---|
| Scene | $-1.00$ | $-0.08$ | $-0.97$ | 1.00 | 1.00 | 1.00 |
| Object | 0.03 | $-0.21$ | $-0.76$ | 0.51 | 0.95 | 0.16 |
| Jigsaw | $-0.74$ | 0.18 | 0.04 | $-1.00$ | $-1.00$ | 1.00 |
| Layout | $-0.65$ | $-0.27$ | 0.57 | $-0.48$ | 1.00 | 0.67 |

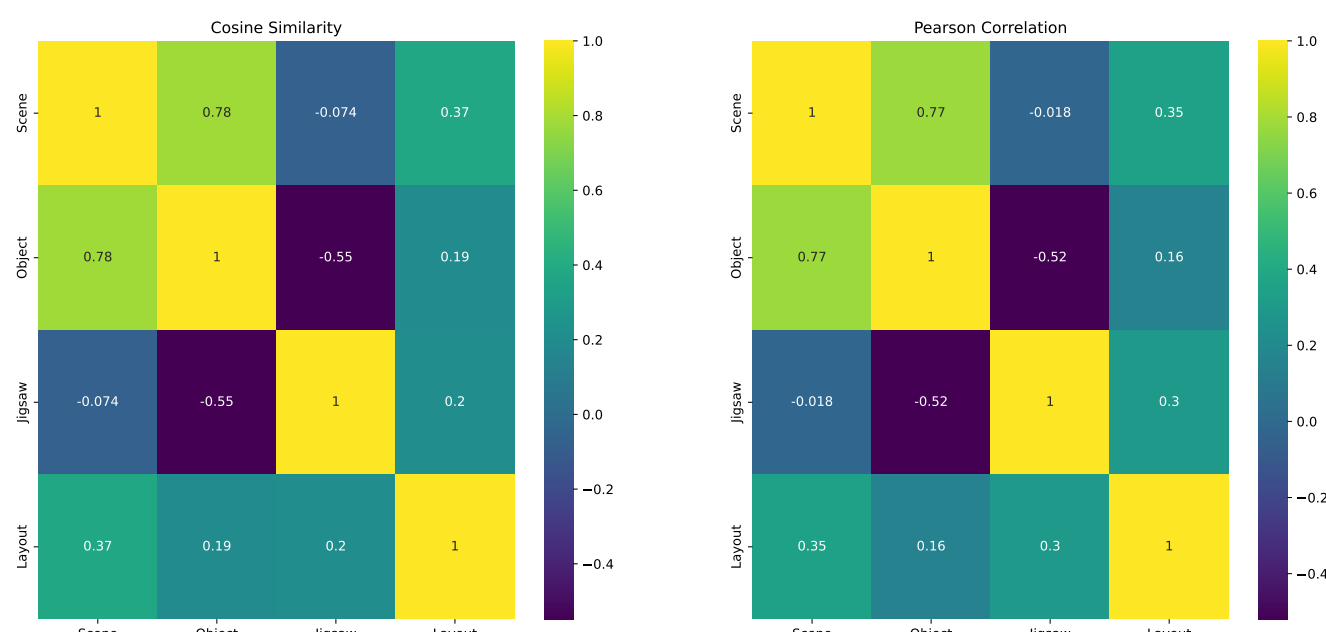

Figure 4: Similarity and correlation among the varying optimized linear combination weights on 4 tasks of TransNAS-Bench-101-micro.

**Ablation Study on Precision @ $T$** To substantiate our claims that the weighted linear combination has better estimation ability and Algorithm 1 can approximate such optimal weight, we demonstrate the Precision @ 100 for the average value of training-free metrics (i.e., $\mathbb{E}[\rho_{\mathrm{T}}(\mathcal{M}, f)]$), the best value of training-free metrics (i.e., $\max(\rho_{\mathrm{T}}(\mathcal{M}, f))$), the optimal value achievable through linear combination (i.e., $\rho^*_{\mathrm{T}}(\mathcal{M}_{\mathbb{M}}, f)$), and the average value of our robust estimation metric (i.e., $\mathbb{E}[\rho_{\mathrm{T}}(\mathcal{M}(\cdot; \widetilde{\boldsymbol{w}}^*), f)]$), as summarized in Table 9. The findings indicate that: *(a)* the linear combination can augment the Precision @ 100, surpassing that of any individual training-free metric, and *(b)* the expected Precision @ 100 value of the robust estimation metric exceeds any single training-free metric and is close to the optimal possible value, indicating that the robust estimation metric has more potential to be exploited.

Table 9: Values of Precision @ 100 of different estimation metrics on TransNAS-Bench-101-micro

| **Estimation Metrics** | **Precision @ 100** | | | | |
|---|---|---|---|---|---|
| | Scene | Object | Jigsaw | Segment | Normal |
| Average | $0.03 \pm 0.03$ | $0.01 \pm 0.02$ | $0.01 \pm 0.01$ | $0.06 \pm 0.05$ | $0.03 \pm 0.01$ |
| Best | 0.08 | 0.05 | 0.01 | 0.14 | 0.04 |
| Optimal | 0.18 | 0.12 | 0.06 | 0.17 | 0.11 |
| RoBoT | $0.14 \pm 0.02$ | $0.09 \pm 0.01$ | $0.04 \pm 0.01$ | $0.15 \pm 0.02$ | $0.09 \pm 0.03$ |

**Ablation Study on $T_0$** As outlined in Section 5.2, to examine the impact of $T_0$, we can experiment with a setting where we strictly prescribe the value of $T_0$. Specifically, with a search budget $T = 100$, we select $T_0$ from [30, 50, 75, 100]. For this setup, we only execute Algorithm 1 for $T_0$ rounds, and we adjust lines 7-10 in Algorithm 1, so now we will always query the architecture that has not yet been queried and holds the lowest ranking value in $\mathcal{M}(\cdot; \boldsymbol{w}_t)$. After running BO for $T_0$ rounds, we will apply the greedy search for the top $T - T_0$ architectures in $\mathcal{M}(\cdot; \widetilde{\boldsymbol{w}}^*)$, as specified in Section 4.3.

Figure 5 demonstrates the outcomes of this experiment. The results highlight that when the BO process concludes (i.e., the exploration phase terminates), there's an immediately noticeable improvement compared to extending the search in BO as the greedy search commences (i.e., the exploitation phase kicks off). However, if the estimation metric used by the greedy search does not exhibit enough potential (i.e., the Precision @ 100 value is lower), it may be overtaken by those with greater $T_0$ values. For instance, in the *Scene* task, while $T_0 = 50$ outperforms other search budgets initially, it's eventually eclipsed by $T_0 = 75$ when querying for 100 architectures. Allocating all the budget for exploration (i.e., $T_0 = T$) could potentially yield poorer results, as seen in the *Scene* task. Our original method, RoBoT, typically delivers the best performance, which suggests that reserving the search budget for duplicate queries and applying them in the exploitation phase is a strategic move to enhance performance. Moreover, it boasts the advantage of not having to explicitly stipulate the value of $T_0$ as a hyperparameter—instead, this value is determined by BO itself.

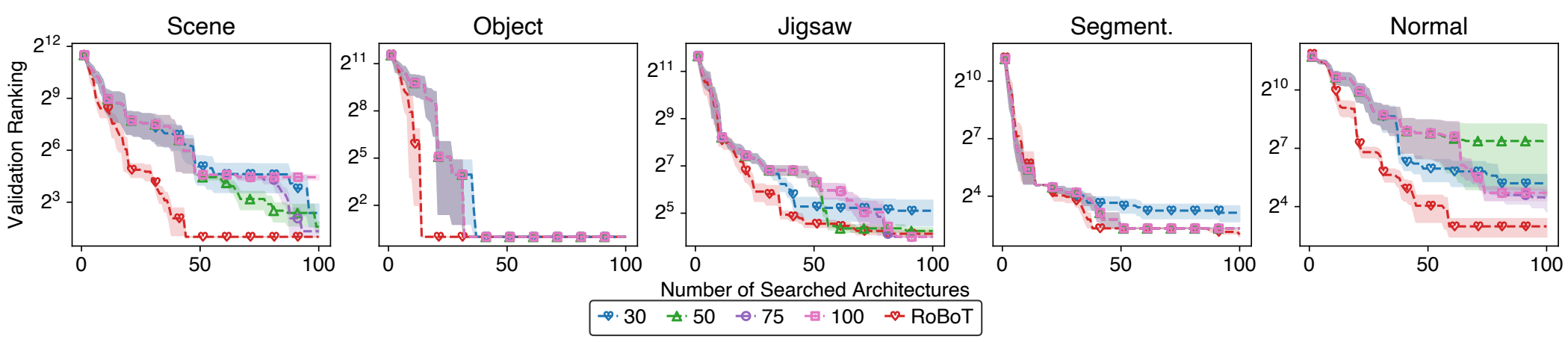

Figure 5: Comparison between different values of $T_0$ and RoBoT on 5 tasks in TransNAS-Bench-101-micro regarding the number of searched architectures. Note that all methods are reported with the mean and standard error of 10 independent searches.

**Ablation Study on Ensemble Method** In this ablation study, we investigate the influence of different ensemble methods on the performance of RoBoT. The original ensemble method in RoBoT involves a weighted linear combination of normalized training-free metrics, where all these metrics collectively form the hypothesis space. To analyze the impact of this ensemble method, we explore alternative approaches through the following ablation studies: *(a) Without Normalization (w/o Normal.):* In this approach, we directly employ a weighted linear combination of the original values of the training-free metrics to generate estimation metrics. *(b) Uniform Distribution (Uniform Dist.):* The values of the training-free metrics are transformed into corresponding ranking values. To maintain the ranking order of architectures, the highest-scoring architecture is assigned a ranking value of $N - 1$ instead of 1 for this method, where $N$ is the number of architectures. These transferred ranking values are then used in the weighted linear combination. As the ranking values are uniformly distributed, we refer to this method as Uniform Distribution. *(c) Normal Distribution (Normal Dist.):* Instead of directly transferring to ranking values, we generate a random normal distribution of $N$ values with a mean of 0 and a standard deviation of 1. These values are then sorted and assigned to the corresponding architectures based on their rankings. It's important to note that the ranking of architectures remains unchanged for all the transferred training-free metrics from the same training-free metric. Thus, all the transferred training-free metrics (from the same training-free metric) have the same Spearman's rank correlation and Kendall rank correlation with the objective evaluation metric.

The results, as shown in Figure 6, indicate several key observations. Firstly, compared to using the original values without normalization, the proposed RoBoT demonstrates a faster convergence in finding the optimal weight vector, resulting in a smaller value of $q_T$ in Theorem 2. This can be attributed to the fact that different training-free metrics often have significantly different magnitudes, and normalization accelerates the optimization process within the BO framework. Secondly, when compared to the Uniform Distribution and Normal Distribution methods that only consider

the rankings instead of the original absolute values of the training-free metrics, RoBoT consistently outperforms them. This observation is interesting since the NAS community often relies on Spearman's rank correlation and Kendall's rank correlation to assess the quality of a training-free metric. However, in this study, we find that all transferred training-free metrics have the same correlation with the objective evaluation metric, yet yield significantly different performances. This suggests that while the absolute values may not be crucial when using a single training-free metric in NAS, they play a vital role in combining multiple training-free metrics. This finding suggests the existence of untapped potential in leveraging the absolute values for training-free metric combinations, and we encourage further investigation by the research community. Overall, the proposed ensemble method RoBoT, which involves a weighted linear combination of normalized training-free metrics, consistently achieves the best performance among the alternative approaches considered in the ablation study.

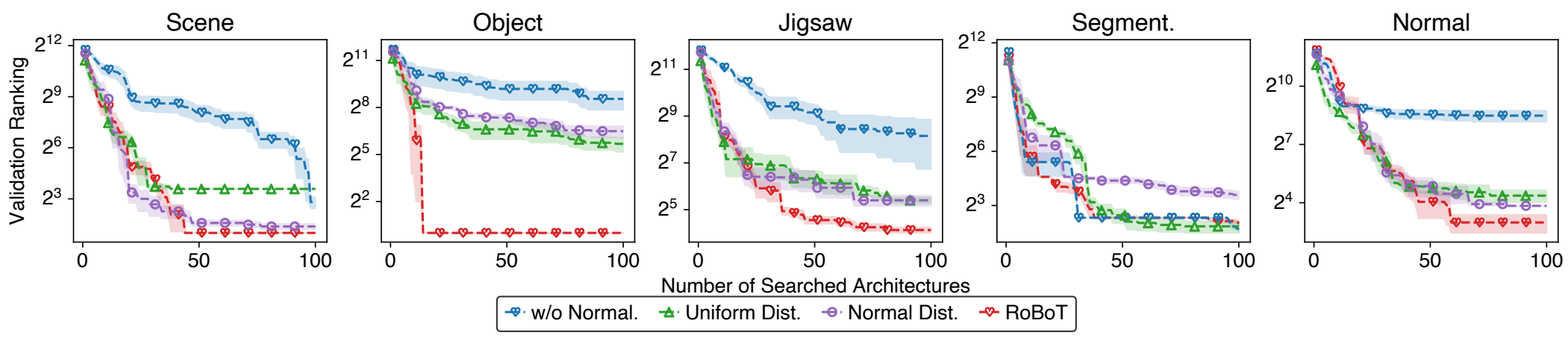

Figure 6: Comparison between different ensemble methods and RoBoT on 5 tasks in TransNAS-Bench-101-micro regarding the number of searched architectures. Note that all methods are reported with the mean and standard error of 10 independent searches.

**Ablation Study on Utilized Training-Free Metrics** In this ablation study, we examine the impact of the training-free metrics used in the linear combination. We consider two scenarios: *(a) More* training-free metrics, where we include *params* and *flops* as additional metrics. This results in a total of 8 training-free metrics for tasks *Scene*, *Object*, and *Jigsaw*, and 7 metrics for tasks *Segment.* and *Normal*. Please note that *synflow* is not applicable for the latter two tasks, as explained in Appendix B.2. *(b) Less* training-free metrics, where we only utilize *grad_norm*, *snip*, and *grasp* for estimation purposes.

The results depicted in Figure 7 demonstrate interesting findings. When utilizing more training-free metrics, the convergence to the optimal weight vector may take longer (as observed in tasks *Scene* and *Object*), but it has the potential to achieve superior performance (as observed in tasks *Jigsaw*, *Segment.*, and *Normal*). The longer convergence time can be attributed to the richer hypothesis space resulting from the inclusion of more training-free metrics. Moreover, the ability to achieve a higher optimum Precision @ $T$ value $\rho^*_{\mathrm{T}}(\mathcal{M}_{\mathbb{M}}, f)$ is also increased. On the other hand, using fewer training-free metrics generally leads to poorer performance, suggesting that the selected training-free metrics may not perform well in the given task. In practical scenarios where prior knowledge about training-free metric performance is limited, it is recommended to include a broader range of training-free metrics for combination.

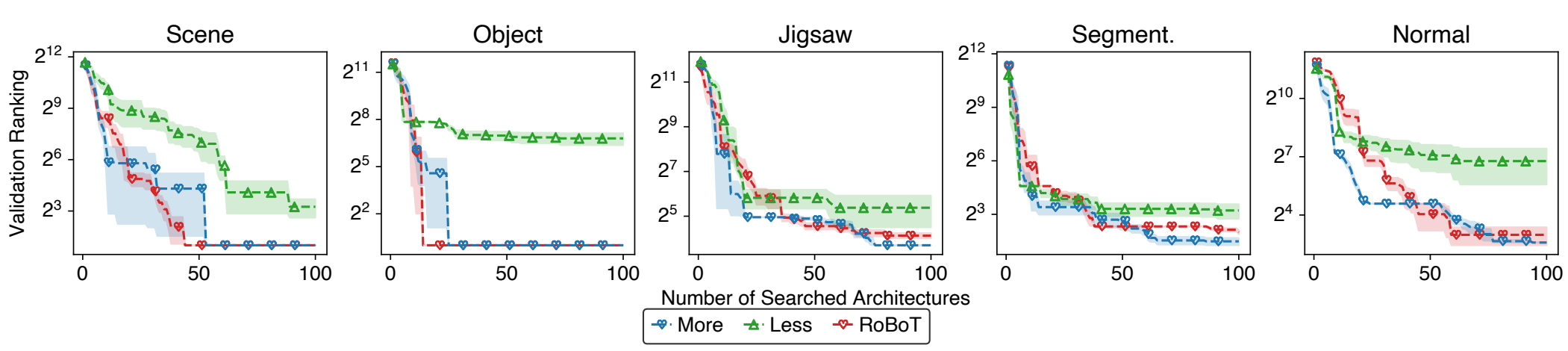

Figure 7: Comparison between utilizing more or less training-free metrics on 5 tasks in TransNAS-Bench-101-micro regarding the number of searched architectures. Note that all methods are reported with the mean and standard error of 10 independent searches.

**Ablation Study on Observation in Algorithm 1** In Section 5.1, we discussed the utilization of the performance of the highest-scoring architecture $\mathcal{A}(\boldsymbol{w}_t)$ as the observation in Algorithm 1 and

emphasized its role as a partial monitoring of Precision @ $T$. This raises the question of whether directly observing Precision @ $T$ would yield superior results. To explore this, we conduct this ablation study.

The findings, presented in Figure 8, clearly indicate that directly observing Precision @ $T$ outperforms the RoBoT approach. This supports our claim about the partial monitoring nature of using the highest-scoring architecture's performance. However, it's important to note that this ablation study is purely hypothetical since practical implementation requires having the performance of all architectures beforehand to compute the Precision @ $T$ value. Nonetheless, this study provides valuable insights into the observation mechanism employed in Algorithm 1 and highlights the practicality and effectiveness of using the performance of the highest-scoring architecture as partial monitoring of Precision @ $T$.

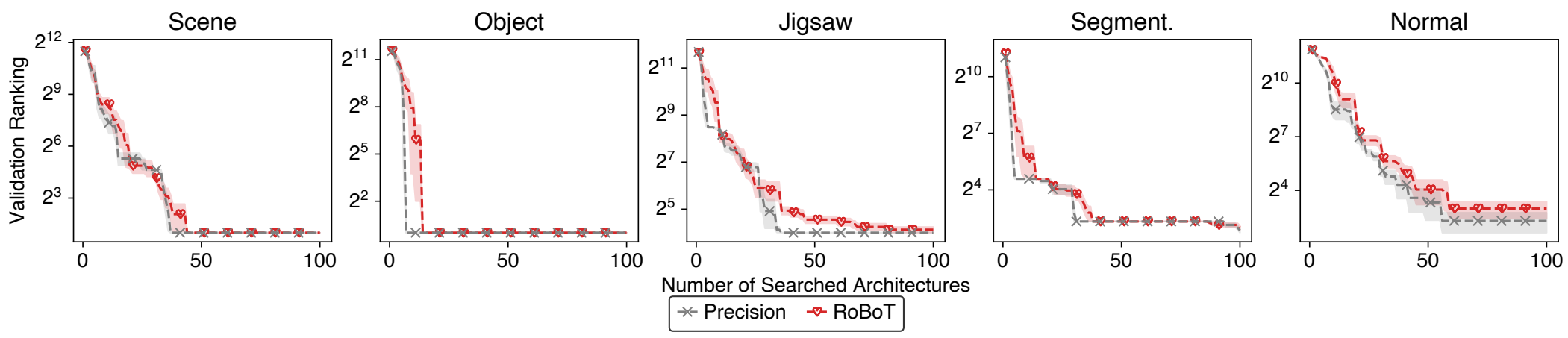

Figure 8: Comparison between directly using Precision @ 100 as observation and RoBoT on 5 tasks in TransNAS-Bench-101-micro regarding the number of searched architectures. Note that all methods are reported with the mean and standard error of 10 independent searches.

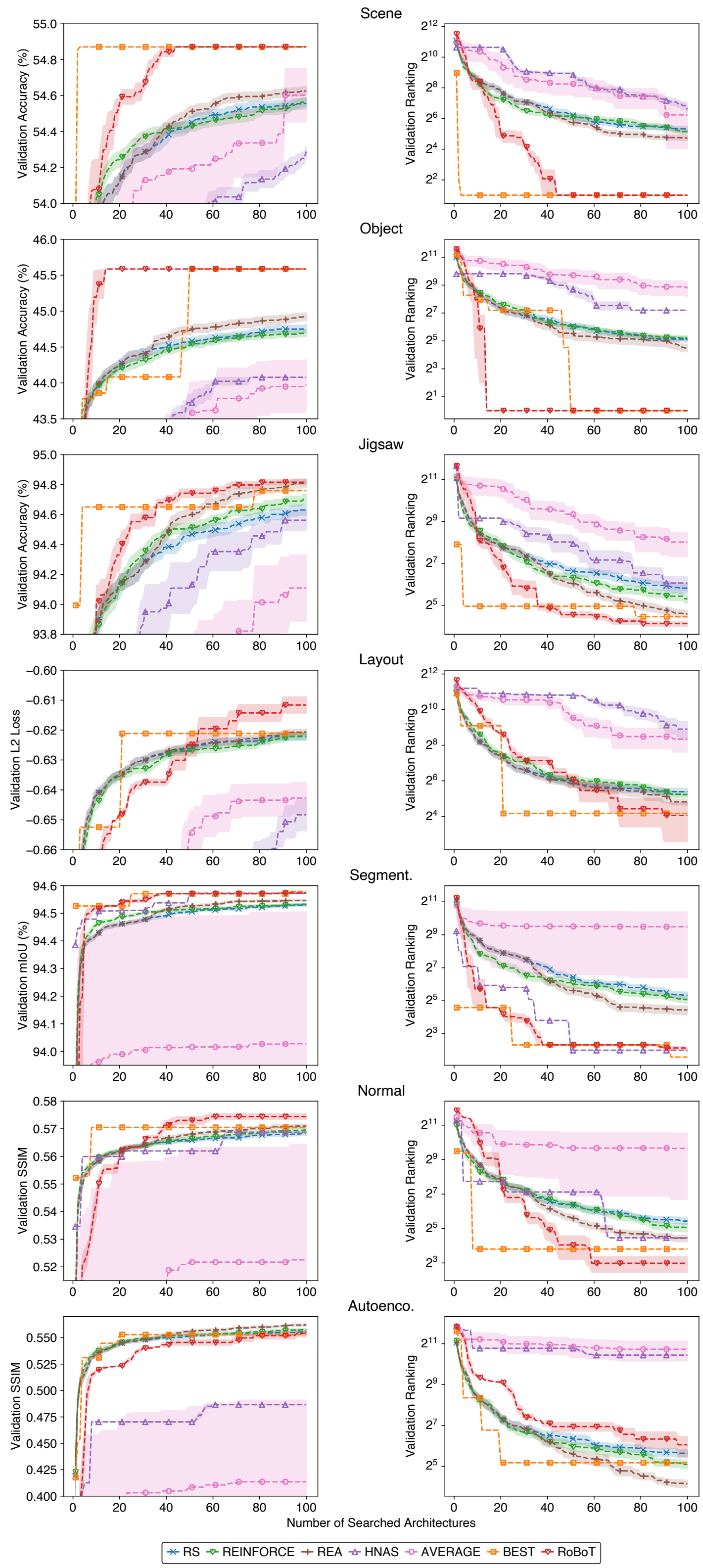

Figure 9: Comparison between RoBoT and other NAS baselines in TransNAS-Bench-101-micro regarding the number of searched architectures. Note that RoBoT and HNAS are reported with the mean and standard error of 10 independent searches, while RS, REA, and REINFORCE are reported with 50 independent searches.

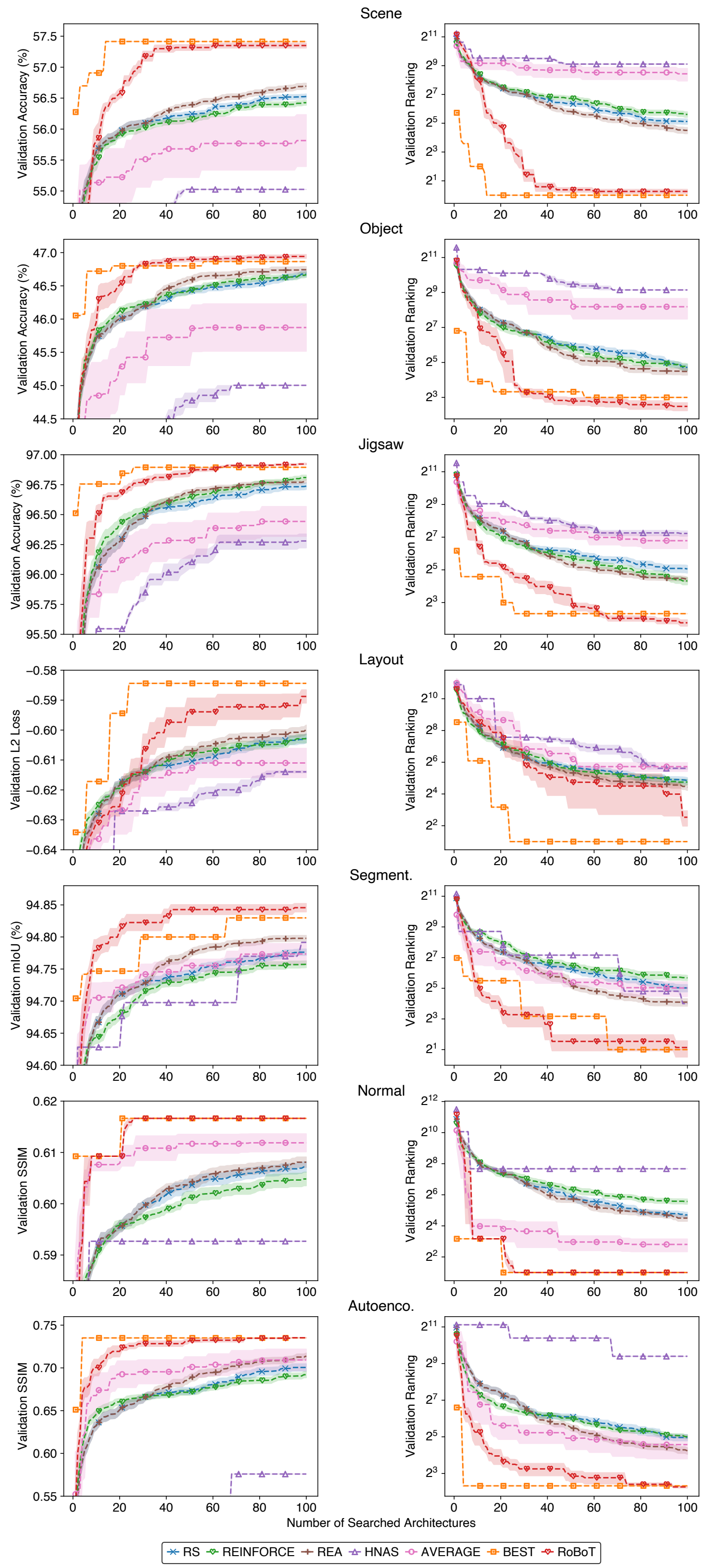

Figure 10: Comparison between RoBoT and other NAS baselines in TransNAS-Bench-101-macro regarding the number of searched architectures. Note that RoBoT and HNAS are reported with the mean and standard error of 10 independent searches, while RS, REA and REINFORCE are reported with 50 independent searches.