# OpenReview forum: "Robustifying and Boosting Training-Free Neural Architecture Search"
_ICLR.cc/2024/Conference — ICLR 2024 poster_

### Official Review · Reviewer_81X4 · 2023-11-01

**Soundness:** 2 fair
**Presentation:** 3 good
**Contribution:** 3 good
**Rating:** 5
**Confidence:** 4

**Summary:**

This work proposes to find a linear combination of training-free metrics to boost the performance on NAS tasks. Specifically, the authors first train a GP to capture the relationship between weights of training-free metrics and the objective evaluation metric f and obtain a robust estimation metric $M^*$. Then the authors collect the queries during the training procedure of BO as $Q_T$. Finally, the authors utilize the learned $M^*$ as a performance estimator and adopt the greedy search to obtain the best architecture.

**Strengths:**

1.	The motivation, that using a linear combination of existing training-free metrics to obtain a robust estimation metric $M^*$, makes sense.

2.	Experiments on NAS benchmarks show the effectiveness of the proposed method.

**Weaknesses:**

1.	The authors propose to train a BO to capture the relationship between the weight vector and the objective evaluation metric f. However, the queried architecture should be trained from scratch to obtain the objective evaluation during the BO stage, which seems to require large amounts of search costs since a standard BO procedure usually requires tens of queries.

BTW: What does $R_f(A)$ denote in Eq. 1? Does it represent the objective evaluation of an architecture? Since Alg.1 directly uses $f$ to denote the objective evaluation metric, I suggest the authors utilize the same notation.

2.	I wonder about the effectiveness of the searched robust estimation metric $M^*$. According to Fig. 2, it seems that the optimal architecture has been found in less than 10 queries during the BO procedure. It shows that there is no need to conduct the greedy search through $M^*$, and BO is enough to get the optimal architecture.

3.	Table 4 shows that RoBoT only requires 0.6 GPU-day to search, does it only count the search cost of the greedy search procedure? I wonder what is the cost of the BO stage, which I am afraid is much larger.

**Questions:**

Please see the weakness.

---

> ### Author Response · Authors · 2023-11-14
> **Response to Reviewer 81X4**
>
> Dear Reviewer 81X4,
>
> We thank you for taking the time to review our paper and for your valuable feedback. We would like to address your concerns below.
>
> ## About the Efficiency of RoBoT
>
> > the queried architecture should be trained from scratch to obtain the objective evaluation during the BO stage, which seems to require large amounts of search costs since a standard BO procedure usually requires tens of queries.
>
> Thank you for pointing out the concern regarding the search efficiency of each query in our method. In practice, we usually apply the validation performance of each candidate architecture that is trained from scratch for only a limited number of epochs (e.g., 12 epochs in NAS-Bench-201) to approximate the true performance of this architecture as stated in our Appendix B.2 and C.2, which in fact is quite computationally efficient (e.g., about 110 GPU-seconds for the model training of each candidate in NAS-Bench-201). Of note, such an approximation, which has been applied in the literature [1], is already reasonably good to help find well-performing architectures, as supported by the results in both Table 2 and Table 4. We would like to add this discussion to the main body of our revised paper to make it clearer.
>
>
> >  RoBoT only requires 0.6 GPU-day to search, does it only count the search cost of the greedy search procedure? I wonder what is the cost of the BO stage.
>
> We thank you for pointing out this question. We would like to clarify that the search cost in our paper includes the computational costs incurred not only in the greedy search stage but also in the BO stage. In fact, these two stages have the same type of training cost that is used to evaluate the validation performance of each queried candidate architecture. Specifically, such a performance evaluation procedure requires about 0.04 GPU-day for the model training of each queried architecture in the DARTS search space for ImageNet where we have queried approximately 3 architectures in the BO stage and 7 architectures in the greedy search procedure. We would like to add this discussion to the main body of our revised paper to make it clearer.
>
>
> ## About the Effectiveness of RoBoT and Greedy Search
> >  According to Fig. 2, it seems that the optimal architecture has been found in less than 10 queries during the BO procedure. It shows that there is no need to conduct the greedy search through $M^*$, and BO is enough to get the optimal architecture.
>
> We would like to clarify that the search costs in Figure 2 include the queries from both the BO stage (e.g., the first 5 queries in Figure 2a that are automatically determined by our RoBoT) and the greedy search procedure (e.g., the last 15 queries in Figure 2a). The results show that greedy search is indeed essential and necessary to help our method achieve competitive search performances, which is further supported by our ablation study in Figure 6. To enhance clarity, we will revise Figure 2 to highlight the search costs incurred by the BO stage and the greedy search procedure separately.
>
>
> ## About Notations
> > What does $R_f(A)$ denote in Eq. 1? Does it represent the objective evaluation of an architecture? Since Alg.1 directly uses $f$
>  to denote the objective evaluation metric, I suggest the authors utilize the same notation.
>
> Thank you for addressing the concern regarding our use of the notations $R_f(A)$ and $f$. As presented in the first paragraph of Section 4, $f(A)$ denotes the objective evaluation metric for $A$, while $R_f(A)$ denotes the **ranking** of $A$ in the entire search space based on the objective evaluation metric $f$. Particularly, $R_f(A)=1$ indicates that architecture $A$ is an optimal architecture in the entire search space based on the evaluation metric $f$. Of note, we introduce $R_f$ mainly to facilitate our theoretical analysis detailed in Section 4.3 and Section 5, given that our theoretical analysis mainly focuses on the ranking performance. But in fact, finding the minimum of $R_f(A(w))$ w.r.t $A(w)$ is essentially equivalent to find the maximum of $f(A(w))$ w.r.t $A(w)$, according to our definition of $R_f(A)=1$.
>
> We agree with you that using both $R_f(A)$ and $f$ in our formulation and algorithm could be confusing. Therefore, we would like to follow your suggestion to revise Equation 1 using only $f$ and reserve $R_f(A)$ only for the theoretical sections. This change aims to make our presentation more coherent and accessible to our readers.
>
>
>  ---
>  Thank you again for your detailed and valuable feedback. We hope that our additional clarifications could help improve your opinion of our paper.
>
>  [1] Unifying and Boosting Gradient-Based Training-Free Neural Architecture Search, NeurIPS 2022

---

> ### Comment · Area_Chair_nYz9 · 2023-11-15
> **Comment on authors' rebuttal?**
>
> @Reviewer 81X4: Does the reply by the authors address the issues raised by you? Do you have any follow-up questions or comments?

---

> ### Author Response · Authors · 2023-11-21
> **Thanks to Reviewer 81X4**
>
> Dear Reviewer 81X4,
>
> We would like to thank you again for appreciating the motivation and effectiveness of our paper and for the constructive comments to further improve our paper. Kindly let us know if you have any further comments on our paper, and we would like to do our best to address them in the remaining time.

---

### Official Review · Reviewer_GJPv · 2023-11-05

**Soundness:** 3 good
**Presentation:** 1 poor
**Contribution:** 2 fair
**Rating:** 3
**Confidence:** 4

**Summary:**

This paper aims to tackle the research gap, the difference between training-free metrics with the final performance. This paper however, propose a weighted linear combination of traditional training free metrics as a estimator, where the weights are obtained automatically via Baysian optimization. Interestingly, this work use partial monitoring theory to prove their method has theoretical performance guarantee. Experiments are conducted on NASBench201.

**Strengths:**

Propose theory seems interesting

**Weaknesses:**

This paper does not read like an academic paper, where the introduction did not cover the full story. Their related work is quite short to cover the existing literature. I suggest the authors try to read more papers in this field instead of submitting their paper in a rush. Results on NASBench201 shows an incremental improvement without realistic benchmarking their method's performance.

**Questions:**

I found the author regularily let the reader "see later section" in their introduction, including Section 3 and section 4. I think this is a not a professional way to write the introduction. We should at least grasp the main idea when reading the intro but instead reading the entire paper.

---

> ### Comment · Area_Chair_nYz9 · 2023-11-15
> **Follow-up**
>
> @Reviewer GJPv: Thanks for your earlier review. Apart from the issues you raised with writing, did you have more specific issues with the method or experiments? If yes, has the revision by the authors improved on any of those issues? If no, could you please list a few pressing issues that still persist, in your opinion? Thanks.

---

### Official Review · Reviewer_3buk · 2023-11-06

**Soundness:** 3 good
**Presentation:** 3 good
**Contribution:** 3 good
**Rating:** 8
**Confidence:** 4

**Summary:**

This work introduces RoBoT, an algorithm for robustifying and boosting training-free neural architecture search (NAS). Motivated by the inconsistent performance estimation of existing training-free NAS metrics, this work proposes to explore a linear combination of multiple metrics that is more robust than each single metric, and exploit the robustified metric combination with more search budgets. The overall framework includes two stages. The first exploration stage employs Bayesian optimization (BO) to find the best linear combination weights for the robust metric. Then, in the second exploitation stage, the remaining search budgets are used to investigate the top-scoring architectures given by the robust metric. The proposed algorithm, RoBoT, is supported by both theoretical and empirical results.

**Strengths:**

- This work is built on existing training-free NAS methods, and extends them to a robustified ensemble. Therefore, the proposed framework is promising for future extension when better training-free NAS methods are discovered.

- Theoretical analysis is provided to understand the proposed algorithm, RoBoT.

- Extensive and solid experiment results on various datasets and settings are provided to demonstrate the efficacy of RoBoT.

**Weaknesses:**

- Missing details regarding robust metric: It seems that some important details about the BO-searched robust estimation metric are missing. What are the base training-free metrics considered in the search? What are the optimized linear combination weights for them? Do they significantly differ on different datasets/tasks?

- Recent NAS methods: It is suggested to include some more recent NAS methods into the comparison, e.g., Shapley-NAS [1], $\beta$-DARTS [2].

Disclaimer: Although I know BO and NAS literature, I’m not familiar with the theoretical background in this work. Therefore, I cannot provide helpful feedback on the theoretical part. I would like to read how other reviewers think about the theoretical results.

[1] Han Xiao, Ziwei Wang, Zheng Zhu, Jie Zhou, Jiwen Lu. Shapley-NAS: Discovering Operation Contribution for Neural Architecture Search. In CVPR, 2022.
[2] Peng Ye, Baopu Li, Yikang Li, Tao Chen, Jiayuan Fan, Wanli Ouyang. $\beta$-DARTS: Beta-Decay Regularization for Differentiable Architecture Search. In CVPR, 2022.

**Questions:**

- In Table 3, why are the results on TransNAS-Bench-101 presented as the validation ranking? It seems to be inconsistent with the accuracy/error in the other two datasets (Tables 2 and 4). Also, the search costs are not listed in Table 3.

---

> ### Author Response · Authors · 2023-11-14
> **Response to Reviewer 3buk**
>
> Dear Reviewer 3buk,
>
> Thank you for recognizing the strengths of our paper and providing the valuable feedbacks. We address your concerns as follows:
>
> ## About Robust Metric
>
> > What are the base training-free metrics considered in the search?
>
> As reported in our Appendix B.2 and C.2, we borrowed the six widely known training-free metrics (i.e., *grad_norm, snip, grasp, fisher, synflow,* and *jacob_cov*) from [1] for most of the experiments in our paper except for the tasks of *Segment.*, *Normal*, and *Autoenco* in TransNAS-Bench-101 where *synflow* was excluded due to its incompatibility with the tanh activation applied in each candidate architecture within the search space.
>
> > What are the optimized linear combination weights for them? Do they significantly differ on different datasets/tasks?
>
> Thank you for raising this interesting question regarding the optimized linear combination weights in various tasks. We present the optimized weights, along with their similarities and correlations, across four tasks in TransNAS-Bench-101-micro as follows:
>
> Varying Optimized Weights
>
> |  | grad_norm | snip | grasp | fisher | synflow | jacob_cov |
> |---|---|---|---|---|---|---|
> | Scene | -1.00 | -0.08 | -0.97 | 1.00 | 1.00 | 1.00 |
> | Object | 0.03 | -0.21 | -0.76 | 0.51 | 0.95 | 0.16 |
> | Jigsaw | -0.74 | 0.18 | 0.04 | -1.00 | -1.00 | 1.00 |
> | Layout | -0.65 | -0.27 | 0.57 | -0.48 | 1.00 | 0.67 |
>
> Cosine Similarity
>
> |  | Scene | Object | Jigsaw | Layout |
> |---|---|---|---|---|
> | Scene | 1.00 | 0.78 | -0.07 | 0.37 |
> | Object | 0.78 | 1.00 | -0.55 | 0.19 |
> | Jigsaw | -0.07 | -0.55 | 1.00 | 0.2 |
> | Layout | 0.37 | 0.19 | 0.2 | 1.00 |
>
> Pearson Correlation
>
> |  | Scene | Object | Jigsaw | Layout |
> |---|---|---|---|---|
> | Scene | 1.00 | 0.77 | -0.02 | 0.35 |
> | Object | 0.77 | 1.00 | -0.52 | 0.16 |
> | Jigsaw | -0.02 | -0.52 | 1.00 | 0.3 |
> | Layout | 0.35 | 0.16 | 0.3 | 1.00 |
>
> The results show that the optimized weights typically vary for different tasks, which aligns with the observations and motivations in Section 3 and further highlights the necessity of developing robust metrics that can perform consistently well on diverse tasks such as our RoBoT. In addition, for tasks with similar characteristics, e.g., the *Scene* and *Object* tasks, both of which are classification tasks, the optimized weights share a relatively high similarity/correlation, indicating the potential transferability of the optimized linear combination within similar tasks. We would like to add these results and discussions to our revised paper.
>
> ## About Including More Baselines
>
> We appreciate your suggestion on more recent baselines. We show our comparison on NAS-Bench-201 as below. Note that we have applied smaller search budgets for Shaply-NAS and $\beta$-DARTS than the ones presented in their original papers in order to maintain comparable search budgets for different training-based NAS algorithms (e.g., DARTS (2nd), GDAS, DrNAS).  The results show that our RoBoT can still achieve better search performances than these two recent baselines.  We will add these baselines and comparisons in our revised paper.
>
> |  | C10 (Test Acc %) | C100 (Test  Acc %) | IN-16 (Test  Acc %) | Cost (GPU Sec.) |
> |---|---|---|---|---|
> | Shaply-NAS | 94.05±0.19 | 73.15±0.26 | 46.25±0.25 | 14762 |
> | $\beta$-DARTS | 94.00±0.22 | 72.91±0.43 | 46.20±0.38 | 3280 |
> | RoBoT | **94.36**±0.00 | **73.51**±0.00 | **46.34**±0.00 | 3051 |
>
> ## About the Presentation of Table 3
>
> >  Why are the results on TransNAS-Bench-101 presented as the validation ranking?
>
> We would like to clarify that since there is a noticeable gap between the validation and the test performances in TransNAS-Bench-101, we follow the common practice [2] to present the validation ranking, as detailed in Appendix B.2.
>
> > the search costs are not listed in Table 3
>
> We would like to clarify that, in fact, we have included the search costs (measured by the number of queries, i.e., 100) in the caption of Table 3.
>
> ---
> We sincerely hope our clarifications above have addressed your concerns and can improve your opinion of our work.
>
>
> [1] Zero-Cost Proxies for Lightweight NAS, ICLR 2021
>
> [2] NAS-Bench-Suite-Zero: Accelerating Research on Zero Cost Proxies, NeurIPS Datasets and Benchmarks Track 2022

---

> > ### Comment · Reviewer_3buk · 2023-11-15
> > **Follow-up on Efficiency**
> >
> > Thank you very much for the helpful response and detailed revision. Most of my previous questions have been addressed.
> >
> > After reading the review from Reviewer 81X4 and your response, I have a quick follow-up question regarding the efficiency of RoBoT. To appoximate the true performance of architectures during the search (both of the BO stage and greedy stage), the model needs to be trained (e.g., "12 epochs in NAS-Bench-201"). I think the model performance after 12 training epochs is already included in NAS-Bench-201. Do you directly retrieve the performance from NAS-Bench-201, or train an actual model and test it? How are the training costs counted?

---

> > > ### Author Response · Authors · 2023-11-16
> > > **Further Clarification on Efficiency**
> > >
> > > Dear Reviewer 3buk,
> > >
> > > Thank you for your follow-up question and we are glad that we have addressed your concerns. We would like to address your follow-up question as follows:
> > >
> > > > Do you directly retrieve the performance from NAS-Bench-201, or train an actual model and test it? How are the training costs counted?
> > >
> > >
> > > We wish to clarify that, as detailed in Appendix B.2, we directly retrieve the performance and corresponding search costs after 12 training epochs (i.e., "hp=12") from NAS-Bench-201 tabular data for each architecture we queried during BO and greedy search stages,  and report the accumulated search costs for all architectures we have queried.
> > >
> > > We truly appreciate your insights and timely feedback. We are more than willing to provide further clarifications or details to address any remaining concerns you might have. Please feel free to share them with us.

---

> > > > ### Comment · Reviewer_3buk · 2023-11-21
> > > > **Reviewer Response**
> > > >
> > > > The authors' responses are appreciated. They have addressed my concerns. I would like to raise my rating.

---

> > > > > ### Author Response · Authors · 2023-11-21
> > > > > **Thank you!**
> > > > >
> > > > > Dear Review 3buk,
> > > > >
> > > > > We thank you very much for reviewing our response and updating the score. We are glad that our response has addressed your concerns and appreciate your recognition of our paper.

---

> ### Comment · Area_Chair_nYz9 · 2023-11-15
> **Comment on authors' rebuttal?**
>
> @Reviewer 3buk: Does the reply by the authors address the issues raised by you? Do you have any follow-up questions or comments?

---

### Author Response · Authors · 2023-11-14
**General Author Response**

Dear reviewers and AC,

We have now addressed all of the suggestions and concerns mentioned by the reviewers. We thank the reviewers very much for these comments, which we believe have substantially improved our work.

The full list of changes in the new version of our paper are as follows

* We rewrite Equation 1 to enhance clarity and coherence for our readers.
* We explicitly specify the information about the utilized training-free metrics in the main body of the paper.
* We add a new section in Appendix B.2 to thoroughly discuss the reported search costs for RoBoT.
* We add and compare two new NAS baselines in NAS-Bench-201.
* We update Figure 2 to more effectively visualize the BO and greedy search stages, aiming to provide clearer insights into our method, RoBoT.
* We add a new section in Appendix C.3 to discuss the varying optimized linear combination weights in different tasks and the similarity/correlation among them, offering additional insights into RoBoT.

We would also like to thank for reviewers for recognizing our paper for the following aspects:
* We have **clear and reasonable** motivation to propose the method RoBoT (Reviewer `81X4`).
* We have conducted **extensive and solid** experiments to demonstrate the **effectiveness** of the proposed method RoBoT (Reviewers `3buk`, `81X4`).
* We have interesting **theoretical analysis** to provide insights for RoBoT  (Reviewers `3buk`, `GJPv`).
* Our method is **promising and flexible** for future extension (Reviewer `3buk`).

---

We thank all reviewers once again for the suggestions and recognition of our papers. We are happy to address any new follow-ups or concerns.

---

### Meta-Review · Area_Chair_nYz9 · 2023-12-13

**Metareview:**

The authors propose a method for robustifying and boosting training-free neural architecture search.  They show that no single training-free metric can consistently outperform each other, thereby creating a need for a more comprehensive metric that is better aligned with accuracy on diverse tasks.
The authors propose to use a weighted linear combination of training-free metrics, which is optimized using using Bayesian optimization, to propose a new metric with better estimation performance.
The reviewers appreciated the simple idea and experimental analysis provided to support the method proposed in the submission. Reviewers raised issues regarding missing details, which were provided in the rebuttal. The authors are strongly encouraged to improve their manuscript based on the reviews and information provided in the rebuttal.

**Justification For Why Not Higher Score:**

The proposed work is not exceptionally groundbreaking and is an incremental, but useful update over prior work. This work still uses a combination of previously proposed network evaluation metrics.

**Justification For Why Not Lower Score:**

Though simple, the proposed work can help improve the quality of training-free NAS methods.

---

### Decision · Program_Chairs · 2024-01-16

Accept (poster)